# Quan-dorcet: Tournament-Based One-vs-One Quantum Classification for Robust Single-Shot Inference

## Abstract

Quantum machine learning (QML) promises powerful classification capabilities, but suffers from fragile output encodings and high sampling demands—especially in multiclass settings. Traditional schemes such as one-hot and binary encoding either produce interpretable outputs too rarely or require many shots to achieve reliable predictions. We propose a decision aggregation framework for quantum multiclass classification based on round-robin tournament scoring. Each output qubit represents a binary comparison between class pairs, and the final prediction is determined by majority wins—yielding a Condorcet-style winner when one exists. This structure improves both the resolvability and accuracy of single-shot predictions, outperforming standard encodings under few-shot conditions. Our method retains global entanglement while localizing decision tasks, enabling interpretable inference that remains reliable under intrinsic quantum randomness, without sacrificing expressivity. Empirical results show that this approach achieves high accuracy and interpretability with significantly fewer measurements, suggesting a promising direction for future quantum classifiers.

## 1 Introduction

Quantum machine learning (QML) seeks to harness the unique properties of quantum systems—such as superposition, entanglement, and interference—to perform learning tasks that may be intractable for classical models. A central tool in quantum machine learning is the parameterized quantum circuit (PQC), a variational quantum model that applies trainable quantum gates to optimize a task-specific objective function (Cerezo et al., 2021a; Schuld and Petruccione, 2021). These circuits are often trained using classical optimization techniques, and their outputs are typically interpreted via expectation values or discretized measurements.

Despite the theoretical promise of QML, an inherent sampling bottleneck poses a fundamental challenge for scalable quantum inference and will persist as a key consideration for future applications. Quantum measurements collapse highly-expressive quantum states into binary outcomes, requiring repeated executions of the circuit to draw samples—or shots—to estimate meaningful statistics (Schuld and Petruccione, 2021). This limitation is particularly acute in multiclass classification, where the structure of the output encoding plays a critical role. In one-hot encoding schemes, the proportion of resolvable outputs—i.e., those that correspond to valid class predictions—vanishes exponentially with the number of classes, making inference increasingly unreliable (Chen et al., 2024). Binary encoding schemes, including standard and Gray codes, avoid this combinatorial collapse but suffer from a different issue: individual bits are often noisy and weakly correlated with the true class, leading to poor accuracy unless a large number of shots are used (LaRose and Coyle, 2020). These problems are distinct but linked by a common theme: the difficulty of extracting reliable, discrete decisions from quantum models under limited measurement budgets. Therefore, rather than focusing on expectation values or aggregate statistics across many shots, we examine the quality of individual measurement samples—what we call *single-shot inference* (not to be confused with few-shot regimes referring to small training datasets).

To evaluate performance under these constraints, we introduce the metric of *shot resolvability*, defined as the probability that a single measurement sample yields a valid and unambiguous class prediction.

Figure 1: Contrived example illustrating common inference errors across different PQC output encoding strategies and the robustness of our tournament-based approach.
(a) *Non-contribution*: Standard PQC setup with angular-encoded inputs, a learnable circuit (we test six variants), and Pauli-Z basis measurements. (b) Binary (similarly Gray) encoding demonstrating misclassification–ie. predicting the wrong class. (c) One-hot encoding demonstrating a nonclassifa-cation–ie. predicting no class. (d) Our tournament mapping decomposes multiclass inference into pairwise quantum comparisons, where each output represents a vote between two classes (darker box indicates the chosen class). (e) Final class is determined by tallying votes across all comparisons. As the number of classes increases, the tournament structure introduces redundancy that helps mitigate both misclassification and nonclassification, improving single-shot inference reliability.

We address the challenge of shot resolvability by introducing a decision aggregation framework for quantum multiclass classification. Rather than relying on global output encodings, our method decomposes the classification task into a series of binary comparisons between class pairs. Each comparison is implemented as a binary quantum classifier operating on a shared entangled state. The outputs of these classifiers are aggregated using a round-robin tournament structure, where each class competes against every other, and the final prediction is determined by majority wins (David, 1959). This approach leverages the statistical robustness of binary decisions and the emergent structure of tournament theory, which ensures that as the number of classes grows, the likelihood of a unique Condorcet-style winner converges to unity while being bound below by $50\%$ (Malinovsky and Moon, 2024).

Importantly, this framework does not discard the global coherence of the quantum model. All classifiers operate within the same entangled quantum state, allowing input information to propagate across the full Hilbert space. The aggregation mechanism simply localizes the decision task, enabling interpretable inference that remains reliable under intrinsic quantum randomness, without sacrificing expressivity. Empirical results show that this method significantly improves accuracy under few-shot regimes, with a particular emphasis on single-shot reliability, outperforming traditional encoding schemes in both sample efficiency and decision consistency.

Our results assume idealized, noiseless conditions to isolate algorithmic behavior from hardware-specific noise. Hardware applicability and noise resilience remain open challenges. Furthermore, our method introduces a trade-off between quadratic qubit scaling and exponential sampling cost. We discuss these limitations in Section 5.

This paper makes the following contributions:

- We introduce *shot resolvability* as a key metric for evaluating the reliability of single-shot predictions in quantum classifiers, providing a practical lens for assessing inference quality under limited measurement budgets.

- We propose a novel output encoding for black-box variational quantum classifiers (VQCs) based on round-robin tournament scoring, leveraging the statistical properties of Condorcet-style decision aggregation to improve both resolvability and accuracy.

- We develop a differentiable training procedure for our tournament-based encoding by embedding pairwise class comparisons into a continuous simplex structure, enabling end-to-end optimization via standard backpropagation.

- We present a comprehensive empirical evaluation across multiple circuit architectures and datasets, demonstrating that our method consistently outperforms standard global encoding schemes in single-shot regimes

## 2 RELATED WORK

### 2.1 QUANTUM CLASSIFICATION

Quantum machine learning (QML) has produced a wide range of classification models, including quantum adaptations of support vector machines (Rebentrost et al., 2014), convolutional neural networks (Cong et al., 2019; Bokhan et al., 2022a), and generative models (Benedetti et al., 2019). Many of these rely on hybrid architectures, where a parameterized quantum circuit (PQC) is embedded within a classical pipeline (Chalumuri et al., 2021; Stein et al., 2022; Shi et al., 2023; Liu et al., 2021). While effective in simulation, hybrid models typically depend on expectation values or floating-point outputs, which require extensive sampling.

Recent work has explored direct multiclass classification using PQCs without hybridization (Zhou et al., 2023; Hur et al., 2022; Shen et al., 2024), but these approaches often rely on thresholding or maximum selection over expectation values, which again necessitate high shot counts. Moreover, most prior methods use global output encodings such as one-hot or binary schemes, which suffer from either low resolvability or poor robustness to bit-level noise (Chen et al., 2024; LaRose and Coyle, 2020; Di Matteo et al., 2021). Some recent efforts have explored alternative encodings such as amplitude-based or angle-based schemes (Schuld et al., 2020), but these typically require deeper circuits or more complex post-selection.

### 2.2 OUTPUT ENCODINGS IN QUANTUM CLASSIFICATION

Most prior work in quantum multiclass classification relies on global output encodings such as one-hot, binary, or Gray code representations. One-hot encoding is conceptually simple and widely used in both classical and quantum settings (Bokhan et al., 2022a; Dhara et al., 2024), but suffers from exponentially vanishing resolvability as the number of classes increases, since only $K$ out of $2^K$ bitstrings correspond to valid outputs (Chen et al., 2024). Binary encoding is more qubit-efficient (often fully valid), but is highly sensitive to bit-flip noise and Hamming-distance errors, which can cause semantically large misclassifications from single-bit perturbations (LaRose and Coyle, 2020; Ding et al., 2025). Gray code encoding mitigates some of this sensitivity by ensuring adjacent class labels differ by only one bit, and has been used in quantum classification tasks (Di Matteo et al., 2021; Bokhan et al., 2022a), but still lacks semantic structure and remains vulnerable to cumulative noise in few-shot regimes.

To improve single-shot reliability, we draw inspiration from tournament theory. Round-robin tournaments have long been studied as a framework for pairwise comparison and ranking (Zermelo, 1929; David, 1959). Recent results show that the probability of a unique Condorcet-style winner in a random tournament converges to unity as the number of classes increases (Malinovsky and Moon, 2024). We leverage this structure to design a decision aggregation framework in which each output qubit represents a binary comparison between class pairs, and the final prediction is determined by majority wins.

## 3 METHOD

We will first introduce the challenges with using existing output encodings for single-shot inference in Section 3.1. Following this, we will present the theory which leads to the improvement in inference of our tournament method over the baselines in Section 3.2. We will then introduce the post-processing method used to differentiably train a PQC to output quality round-robin tournament results in Section 3.3, as well as other training decisions. After the main contributions, we discuss the circuit setup and variations in Section 3.4 and the computational tools used in Section 3.5. These sections are included to increase reproducibility, however for the interested reader, we have expanded upon the intricacies of the the circuit blocks we choose in Section A.3 and on quantum computation in general in Section A.4 .

### 3.1 OUTPUT ENCODING EFFECTS ON SHOT RESOLVABILITY

In quantum multiclass classification, the choice of output encoding plays a critical role in determining both the resolvability of measurement outcomes and the accuracy of predictions under limited

sampling. We evaluate four encoding strategies—one-hot, binary, Gray code, and our proposed tournament-based encoding—with a focus on their behavior under few-shot and single-shot inference regimes.

*One-Hot* - One-hot encoding assigns each class to a unique qubit, with the correct class represented by a single qubit in the excited state (e.g., $|1\rangle$) and all others in the ground state ($|0\rangle$). This encoding is conceptually simple and widely used in classical multiclass classification. However, in quantum settings, it suffers from a severe validity bottleneck: only $K$ out of $2^K$ possible bitstrings correspond to valid one-hot outputs, where $K$ is the number of classes. Thus, the probability of obtaining a resolvable output from a random measurement decays exponentially as $P_{\text{valid}} = K/2^K$, making inference unreliable under few-shot conditions (Chen et al., 2024).

*Binary and Gray Code Encoding* - Both binary encodings map each class label to a binary representation across $\lceil \log_2 K \rceil$ qubits. These encodings are highly efficient in terms of qubit usage and have maximal resolvability: every bitstring corresponds to a class label, modulo padding for non-power-of-two class counts. However, robustness to sampling variability under few-shot inference is poor. Individual qubits contribute to multiple bits of the class label, and noise in any bit can lead to misclassification. Moreover, binary encoding is sensitive to Hamming distance errors, where small perturbations in the bitstring can result in large semantic shifts in class prediction (LaRose and Coyle, 2020).

Gray code encoding modifies binary encoding such that consecutive class labels differ by only one bit. This reduces the impact of single-bit errors, improving robustness under low-shot conditions. However its accuracy gains under single-shot inference are modest and context-dependent. In our experiments, Gray code fails to unilaterally outperform standard binary encoding in single-shot accuracy, and still falls short of our tournament-based method.

Table 1: Comparison of output encoding strategies in terms of validity and accuracy under few-shot and many-shot regimes.

| Encoding Method | Resolvability | Accuracy (Single-Shot) | Accuracy (Many-Shot) |
|---|---|---|---|
| One-Hot | Low ($\sim K/2^K$) | Low | High |
| Binary | High (Full coverage) | Moderate | Moderate–High |
| Gray Code | High (Full coverage) | Moderate-Low | Moderate–High |
| Tournament (Ours) | High ($\to 1$ as $K \to \infty$) | High | High |

## 3.2 TOURNAMENT-BASED ENCODING (OURS)

Our proposed encoding frames multiclass classification as a round-robin tournament among class pairs (Moon et al., 1968). Each output bit represents a binary decision between two classes, and the final prediction is determined by majority wins (Copeland-style). This structure corresponds to an orientation of a complete directed graph over $K$ vertices, with $K(K-1)/2$ binary decisions. While our method does not require a Condorcet winner to produce a prediction, Condorcet theory provides the probabilistic guarantees that underpin tournament encoding and motivate the Quan-dorcet design.

Theoretical results from Malinovsky and Moon (2024) show that the probability of a unique winner in a random tournament converges to 1 as $K \to \infty$. This implies that even stochastic or partially incorrect binary decisions can yield a resolvable class prediction, meaning the convergence properties remain valid regardless of hardware noise or backend fidelity. In this work, we evaluate the method under idealized, noiseless conditions to isolate algorithmic behavior and demonstrate these properties empirically. As illustrated in Figure 2, even when cycles occur among some pairwise comparisons, the Copeland-style aggregation still produces a unique prediction. The example shows a $K = 4$ tournament where three edges agree and the remaining form a cycle, demonstrating that cycles do not prevent resolvability under our framework.

Unlike one-hot or binary encodings, tournament-based encoding does not require global agreement across qubits. Each decision is localized, yet the model retains global coherence via shared entanglement. This duality of local decision simplicity with global state expressivity is a key factor in the superior performance of the method. Empirically, our results in Section 4.2 and Section 4.3

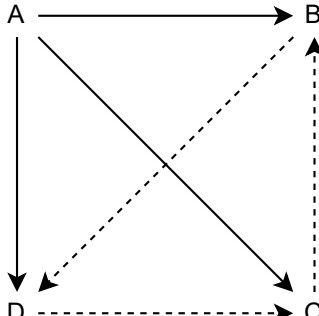

Figure 2: A round-robin tournament of four classes. Arrows indicate the direction of pairwise preference (from the class that wins the comparison to the class that loses). Note the cycle (dashed) among classes B, C, and D—B beats D, D beats C, and C beats B—where no Condorcet winner exists. However, class A defeats all others and is selected as the tournament winner by majority vote (Copeland-style). If A had lost to any one of the other three, the result would be a tie, rendering the tournament unresolvable under our framework.

show our method achieves high resolvability and accuracy even with a single shot, outperforming other encodings in low-shot regimes while matching their performance under high sampling and simulation.

### 3.3 TRAINING PROCEDURES

All training is conducted under noiseless simulation using the same PQC architecture described in Section 3.4. This standard practice in quantum machine learning isolates algorithmic behavior from hardware-specific noise and ensures fair comparison across encodings. Our focus is on how each output encoding is interpreted and optimized during training, giving each method the best opportunity to perform under its own assumptions.

All encodings require a continuous representation for gradient-based optimization. For binary encodings (binary and Gray), this is straightforward: the model outputs continuous values that can be trained using binary cross-entropy or distance-based losses against a known bitstring target. One-hot encoding, widely used in both classical and quantum classification (Bokhan et al., 2022a; Chen et al., 2024), corresponds to moving the center of mass of a probability simplex toward the correct vertex, and is typically trained using cross-entropy loss.

Our tournament-based encoding presents a unique challenge: it produces a vector of binary comparisons between class pairs, many of which are undefined for a given target class. Specifically, only the comparisons involving the true class $c_k$ have a well-defined target; the rest are structurally ambiguous. This makes it inappropriate to apply a bitwise loss across all outputs. To address this, we introduce a novel continuous training method that leverages the geometric correspondence between round-robin tournaments and the edges of a regular simplex. By interpolating each binary comparison along the edge connecting its two associated class vertices, we construct class-specific mass points within the simplex. This allows us to compute distances to the true class vertex and apply a softmax-based loss, analogous to one-hot training.

Formally, the PQC outputs expectation values $\langle Z_i \rangle$ for each qubit $i$, corresponding to binary comparisons between class pairs $(c_i, c_j)$. These are passed through a nonlinear activation function $\phi : [-1, 1] \rightarrow [0, 1]$ (see Section A.1.3) to produce confidence scores $e_{ij} = \phi(\langle Z_{ij} \rangle)$. Each score is used to interpolate between the vertices $v_i$ and $v_j$ of a regular, zero-centered $(K-1)$-simplex:

$$\mathbf{p}_{\{i,j\}} = (1 - e_{\{i,j\}})\mathbf{v}_i + e_{\{i,j\}}\mathbf{v}_j.$$

For each class $c_k$, we compute the average of the interpolated points along its incident edges:

$$\mathbf{n}_k = \frac{1}{K-1} \sum_{j \neq k} \mathbf{p}_{\{k,j\}}.$$

This yields a class-specific mass point $\mathbf{n}_k$ within the simplex. We then compute the Euclidean distance between each class's mass point and its corresponding vertex:

$$d_k = \|\mathbf{n}_k - \mathbf{v}_k\|,$$

and apply a softmax transformation to the inverted distances to produce class scores:

$$p_k = \frac{\exp(1 - d_k)}{\sum_{j=1}^{K} \exp(1 - d_j)}.$$

These scores are used in a symmetric cross-entropy loss:

$$\mathcal{L} = \sum_{k=1}^{K} \left[ y_k \log(p_k) + (1 - y_k) \log(1 - p_k) \right],$$

where $y_k$ is the one-hot target label for class $k$. This formulation retains the benefits of softmax normalization while preserving class-specific optimization manifolds. Unlike standard cross-entropy, which only penalizes incorrect predictions, symmetric cross-entropy encourages confident separation between correct and incorrect classes. This is particularly beneficial in our setting, where each class is defined by its incident binary comparisons. Our observations are consistent with prior work showing that symmetric cross-entropy improves class separation and robustness to sampling variability (Wang et al., 2019; Das and Chaudhuri, 2019; Huang et al., 2020).

One-hot training uses the same softmax symmetric cross-entropy directly on the activations of the expectation values $\langle Z_i \rangle$ from the PQC. For binary and Gray code encodings, we use the same symmetric cross-entropy formulation, omitting the softmax normalization step, as the targets are bitstrings rather than one-hot vectors.

Each PQC model is trained with a batch size of 32 for 6 epochs using the Adam optimizer with an exponential decay learning rate scheduler (Kingma and Ba, 2017), with a starting learning rate of 0.01, a decay rate of 0.9, and scheduler steps equal to one-tenth of the total training steps. This configuration was selected based on an ablation study in Section A.1.4 .

## 3.4 CIRCUIT DESIGN

To ensure consistency and comparability across encoding methods, all experiments use a shared PQC architecture. We adopt the dual-angle encoding scheme from Hur et al. (2022); Munikote (2024), which has demonstrated strong performance in prior work. Input features are encoded using $W = \binom{K}{2}$ qubits, where $K$ is the number of classes. Each qubit receives two features—one via a Pauli-X rotation and one via a Pauli-Y rotation—yielding a total of $2W$ encoded features. Input data is scaled to the range $[-1, 1]$ to ensure unique embeddings, and dimensionality reduction is performed using a reproducible autoencoder with dropout (Bishop, 2006).

The main circuit topology is a 2-design qubit ring (Cerezo et al., 2021b), where each wire is connected to its two neighbors via alternating layers of computational blocks. These blocks consist of parameterized single-qubit rotations and two-qubit controlled operations. We evaluate six well-established block types: CNN7 and CNN8 (Sim et al., 2019; Hur et al., 2022), $SO(4)$ and $SU(4)$ (Wei and Di, 2012; Vatan and Williams, 2004). We also test on a slightly different multi-qubit entangling setup known as Strongly Entangling Layers (Schuld et al., 2020). This setup applies parameterized SU(2) rotations on each individual qubit, then applies a 2-qubit controlled gate to each consecutive pair of qubits. We test this setup with both CNOT and controlled-Z gates as the 2-qubit gates. Descriptions and diagrams of each block are provided in Section A.3 , and a schematic of the overall setup is shown in Figure 1. We use four layers of ring blocks or SEL layers in all experiments, though this depth can be adjusted to trade off expressivity and gate cost, as shown in Section A.1.1 . Importantly, our results are not tied to any specific circuit block or depth—our method operates as a post-processing framework and is compatible with a wide range of architectures.

Measurement strategies differ slightly between encoding methods: the one-hot framework measures $K$ qubits corresponding to class vertices, binary and Gray frameworks measure $\lceil \log_2 K \rceil$, and the tournament framework measures all $W$ qubits. Measuring a subset of wires is standard practice in PQC training (Hur et al., 2022; Bokhan et al., 2022b; Zhou et al., 2023; Shen et al., 2024; Stein et al.,

2022), and has even been linked to improved gradient behavior and reduced barren plateau effects (Cerezo et al., 2021b; Leone et al., 2024; Cerezo et al., 2024). All measurements are performed in the Pauli-Z basis.

### 3.5 COMPUTATIONAL TOOLS

All experiments were performed using the Python packages JAX (Bradbury et al., 2018) and Penny-Lane (Bergholm et al., 2022). JAX is an auto-differentiation package that enables the computation of gradients for machine learning models and just-in-time compilation for highly parallel processes such as batched PQC operations. PennyLane is a superconducting quantum computing package for Python that interfaces with most modern superconducting quantum computer APIs and machine learning packages, including JAX, which enables rapid training and testing of the PQCs used in this work. The full spread of experiments was obtained from 100 kCPU-hours on two Intel Xeon Gold 6130s, with another 300 used for the ablations.

Due to our primary contributions being post-processing methods, our main results are computed under noiseless CPU simulation, though our ablation in Section A.1.2 shows that the relative inference performance between the tournament and one-hot methods changes very little when performed using (retired) IBM noise models. In practice, simulation remains feasible for small $K$ (e.g. $K \leq 6$) and shallow circuits ($\leq 4$ layers), which we report in Section 4. These regimes reflect the intended scope of this study: evaluating encoding strategies under strict sampling constraints rather than optimizing for hardware execution. Code is provided on (Anonymous) GitHub.

## 4 RESULTS

To evaluate the performance of our tournament-based decision aggregation framework, we compare it against one-hot, binary, and Gray code. We train and test on five permutations of two datasets and six block circuits each, using the same random seeds for each permutation for direct comparison of the methods. Tests were done on both the MNIST Digits dataset (LeCun et al., 1998) and the MNIST Fashion dataset (Xiao et al., 2017), with five random subsets of $K$-classes. The same five random subsets were used for each permutation with additional classes chosen from the remaining digits as $K$ increases. These chosen datasets are balanced and have clean labels to isolate encoding effects. Robustness to imbalance and overlapping classes is an open question and discussed in Section 5.

### 4.1 METRICS

We report results using metrics designed to reflect the reliability of discrete predictions under limited sampling. The resolvable accuracy $A_R$ measures the proportion of resolvable measurement shots that yield the correct class label, capturing per-shot correctness. The resolvability ratio $R$ quantifies the fraction of resolvable measurement shots, and is calculated by measuring until 100 resolvable shots are collected, and dividing 100 by the number of shots needed to be measured to acquire those 100.

To assess single-shot performance more directly, we also present the shot accuracy $A_s$, computed over a fixed number of shots per test sample (e.g., 100), including unresolvable predictions. This metric reflects the probability that a single shot yields a correct prediction, and serves as our primary measure of single-shot inference quality. Additionally, we define the effective accuracy $A_e = RA_R$. This metric captures the expected correctness of a randomly sampled shot. While their definition implies $A_s \approx A_e$, our results show empirically that this is not the case, and that there is a correlation between resolvability and accuracy.

### 4.2 RESOLVABILITY

In Table 2, we report the resolvability ratio $R$ and the resolvable-shot accuracy $A_R$. The relationship between the two is plotted on the left part of Figure 3. Tournament encoding consistently achieves high resolvability and strong per-shot correctness across all experiments. In contrast, binary and Gray encodings—despite producing resolvable outputs—exhibit lower accuracy due to their sensitivity to sampling fluctuations and lack of semantic structure in the output space.

Table 2: Comparison of our proposed tournament encoding against common encodings when looking for the ratio of resolvable outputs and their class accuracy. Tournament encoding demonstrates reliable performance over both metrics, while one hot has high accuracy with less resolvability, and the two binary-methods are highly resolvable but less accurate.

| | | $R$ | | | | $A_R$ | | | |
|---|---|---|---|---|---|---|---|---|---|
| | | Tournament | One-Hot | Binary | Gray | Tournament | One-Hot | Binary | Gray |
| Digits | 3 | 93.16% | 57.41% | 92.04% | 90.64% | 58.27% | 60.64% | 54.50% | 54.40% |
| Digits | 4 | 71.11% | 38.70% | 100.00% | 100.00% | 57.55% | 60.25% | 49.04% | 25.05% |
| Digits | 5 | 66.01% | 23.47% | 78.43% | 78.32% | 47.36% | 49.65% | 39.44% | 39.76% |
| Digits | 6 | 67.14% | 15.54% | 83.17% | 83.69% | 42.63% | 41.44% | 31.02% | 31.39% |
| Fashion | 3 | 97.38% | 46.62% | 90.38% | 94.92% | 60.46% | 60.51% | 56.58% | 56.90% |
| Fashion | 4 | 76.24% | 36.55% | 100.00% | 100.00% | 53.28% | 55.16% | 45.68% | 25.22% |
| Fashion | 5 | 71.36% | 24.63% | 81.61% | 81.65% | 48.26% | 50.78% | 41.63% | 42.63% |
| Fashion | 6 | 68.11% | 15.33% | 84.03% | 84.22% | 41.11% | 41.06% | 31.01% | 31.62% |

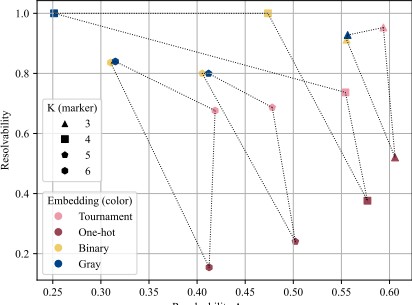
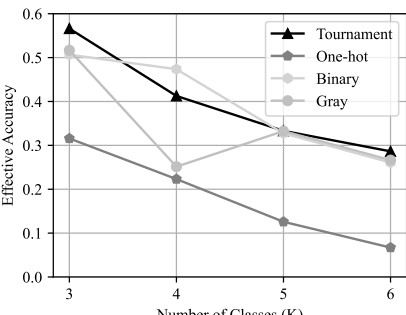

Figure 3: *Left:* Plot showing the spread of resolvability and resolvable accuracy across all methods and $K$. Notice that the tournament method is always in the top-right corner, indicating solid reliability. *Right:* Plot showing trends in effective accuracy as K increases. While calculated from the left plot, clearly shows the consistency of the tournament method as compared to the erratic behavior of binary methods and the poor performance of one-hot.

One-hot encoding suffers from low resolvability, particularly as the number of classes increases. While resolved shots tend to be accurate—indicating that the correct qubit is often excited—many incorrect qubits are also excited simultaneously. This reflects a fundamental limitation of one-hot encoding: it attempts to represent a confidence distribution over classes, but any nonzero confidence in an incorrect class can lead to misclassification. Inference requires extensive sampling to recover the dominant excitation.

These results confirm that tournament encoding uniquely balances resolvability and correctness, yielding interpretable and accurate predictions with minimal sampling. Importantly, this balance is maintained even as $K$ increases, unlike binary encodings which exhibit sharp structural degradation. This stepwise breakdown is a direct consequence of the discrete nature of $\lceil \log_2 K \rceil$ and foreshadows the scaling limitations of binary methods.

### 4.3 Shot Quality Analysis

To further illustrate the trade-offs between encodings, we combine the resolvability metrics into effective accuracy $A_e = RA_R$ and compare it to the measured shot accuracy $A_s$. These metrics are similar when resolvability is high, but diverge as resolvability drops, since $A_s$ includes unresolvable outputs. As shown in Table 3 and the right side of Figure 3, tournament encoding maintains strong performance across all class counts, while one-hot encoding degrades sharply.

Notably, binary encoding performs well at $K = 4$, but its accuracy drops significantly at $K = 5$, coinciding with a steep decline in resolvability. This highlights the importance of resolvability as a metric: binary encodings are structurally bound to degrade as $K$ increases. Gray encoding

exhibits similar behavior, with accuracy falling below random guessing in some cases, suggesting that semantic distance between classes is not preserved under sampling variability.

Table 3: Comparison of our proposed tournament encoding against common encodings when looking at the overall accuracy of discrete PQC accuracies. Shot accuracy and $A_e$ are theoretically equal when resolvability is maximal, leading to a correlation for tournament, binary, and Gray encodings on the (far simpler) Digits dataset. One-hot degrades as resolvability goes down.

| | | $A_s$ | | | | $A_e = RA_R$ | | | |
| --- | --- | --- | --- | --- | --- | --- | --- | --- | --- |
| | | Tournament | One-Hot | Binary | Gray | Tournament | One-Hot | Binary | Gray |
| Digits | 3 | 54.62% | 38.82% | 50.41% | 49.93% | 54.33% | 34.89% | 50.18% | 49.33% |
| Digits | 4 | 42.89% | 27.41% | 49.04% | 25.05% | 41.44% | 24.03% | 49.04% | 25.05% |
| Digits | 5 | 32.60% | 14.86% | 31.72% | 32.01% | 31.68% | 12.26% | 31.26% | 31.44% |
| Digits | 6 | 29.63% | 8.21% | 26.23% | 26.75% | 28.93% | 6.81% | 25.96% | 26.47% |
| Fashion | 3 | 59.03% | 41.59% | 52.27% | 54.49% | 58.90% | 28.26% | 51.15% | 54.02% |
| Fashion | 4 | 41.95% | 25.02% | 45.68% | 25.22% | 41.08% | 20.60% | 45.68% | 25.22% |
| Fashion | 5 | 35.73% | 16.59% | 34.68% | 35.40% | 34.93% | 12.90% | 34.20% | 35.05% |
| Fashion | 6 | 29.05% | 8.28% | 26.49% | 27.13% | 28.34% | 6.59% | 26.19% | 26.79% |

These trends reinforce the practical advantage of tournament encoding: it produces high-quality predictions without requiring filtering or extensive measurement. Unlike binary methods, tournament aggregation does not rely on a fixed bitstring structure and instead leverages pairwise comparisons, which scale more gracefully with $K$. This motivates a deeper analysis of scaling behavior, which we explore in the next section (Section 4.4) by examining the area-under-curve (AuC) of resolvability for each method.

## 4.4 SCALING BEHAVIOR AND STRUCTURAL LIMITS

While our experiments focus on relatively small class counts ($K \leq 6$), the structural implications of each encoding become increasingly important as $K$ grows. Binary and Gray encodings exhibit a discrete-to-exponential mismatch: the number of valid class labels grows linearly with $K$, while the number of possible bitstrings grows exponentially with the number of qubits. This leads to a bounded oscillatory degradation in resolvability, particularly when $\log_2 K$ is not an integer. For example, binary encoding achieves full validity at $K = 4$ (using 2 qubits), but drops to $62.5\%$ validity at $K = 5$ (using 3 qubits), as only 5 of the 8 possible bitstrings correspond to valid class labels. This structural fragility implies that binary encodings are inherently sensitive to class count and qubit budget. For non-power-of-two $K$, the fraction of valid bitstrings decreases, and the probability of generating an unresolvable output rises sharply. This behavior is not merely empirical—it is a direct consequence of the encoding scheme's discrete nature. We present the theoretical lower bounds for the resolvability probability of random bitstrings in Figure 4.

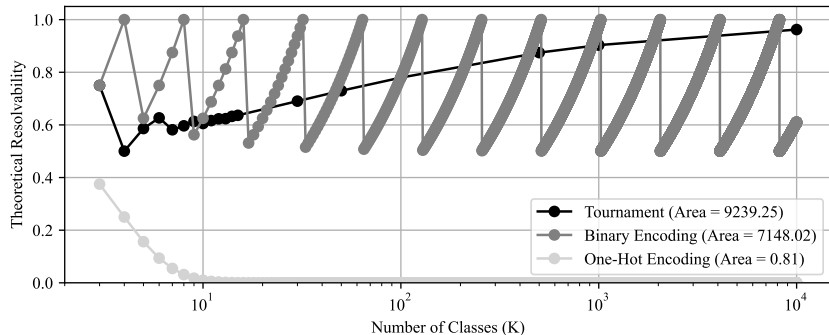

Figure 4: Figure comparing the $K \rightarrow \infty$ resolvability of one-hot encoding, both binary encodings, and our tournament encoding. Areas shown the legend account for total area under the resolvability curve, allowing for fair comparison between the oscillating binary curve and the smooth tournament and one-hot curves. Note that the plots start at $K = 3$.

Moreover, the aforementioned results by Malinovsky and Moon (2024) show that the likelihood of Condorcet-style aggregation converges to unity as $K \rightarrow \infty$. This convergence property is absent in binary encodings, for which there is an oscillatory bounded curve for all $K$. For one-hot, we use $K/2^K$, which decreases exponentially until vanishing.

## 5 LIMITATIONS

Our proposed tournament-based encoding introduces a fundamental trade-off: quadratic qubit scaling with respect to the number of classes $K$. This requirement makes the approach impractical for large-scale problems until quantum hardware achieves significant improvements in qubit availability and fidelity. Consequently, all results in this paper are obtained under idealized conditions to isolate algorithmic behavior from hardware-specific noise. While this choice enables a clear evaluation of encoding strategies, robustness to real-device imperfections and resource constraints—both general and tournament-specific—remains an open challenge.

These constraints position our work as a theoretical analysis of output encodings rather than a direct path to near-term hardware deployment. The guarantees we provide, such as the convergence of resolvability to unity as $K \rightarrow \infty$ (Malinovsky and Moon, 2024), are purely combinatorial and hold regardless of backend fidelity or noise. Our empirical evaluation under noiseless simulation demonstrates these properties in practice, but does not claim hardware readiness.

In addition, our experiments assume balanced datasets with clean labels. Class imbalance and semantic overlap introduce structural challenges: imbalance may bias majority voting toward dominant classes, while overlapping decision boundaries can increase the likelihood of cycles, which our current framework discards as "unresolvable." These effects are not unique to quantum classifiers—they also affect classical one-vs-one schemes—but their impact on resolvability and accuracy under tournament aggregation remains an open question. We highlight these limitations explicitly and view extensions such as weighted voting, cycle-aware heuristics, and adaptive tie-breaking as promising directions for future work.

Future research should also explore strategies to mitigate quadratic scaling, such as hierarchical or sparse tournament structures, hybrid aggregation schemes, and alternative scoring mechanisms. Extensions inspired by classical tournament theory (e.g., Condorcet-cycle handling, Schulze methods) offer promising directions for improving both efficiency and resolvability. Assuming continued progress in quantum hardware and deeper theoretical development, a large-scale experimental study on real quantum processors would be a natural next step. Such work is essential before deploying Quan-dorcet-style models on high-dimensional datasets or production-level tasks.

Finally, we acknowledge that a language model was used to refine the clarity and consistency of the manuscript. All conceptual contributions, experimental design, and theoretical insights remain entirely our own.

## 6 CONCLUSIONS

In this paper, we take the first step toward improving the resolvability and accuracy of discrete outputs from multi-class PQC classifiers. Achieving this goal has broader implications for quantum machine learning, as reducing sampling requirements removes a significant obstacle to quantum computing. Our findings, supported by experiments, highlight a novel direction in quantum machine learning. We focus on designing models that yield resolvable and accurate discrete outputs more often by leveraging tournament solutions.

To achieve this, we propose a classical post-processing method for PQCs that maps the output space to a regular simplex, leading to the model learning a probabilistic directed graph over classes. Under deterministic inference, such models produce resolvable samples whenever there is a unique majority "winner," rather than only when a an exact bitstring is produced. This effect upper bounds sampling needs as the number of classes increases while still producing highly accurate single-shot measurements, as compared to stand one-hot or bitstring based methods.

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

# A APPENDIX

## A.1 ABLATIONS

### A.1.1 CIRCUIT DEPTH

In order to test the assumption that the results of our post-processing method are unrelated to circuit depth, we ran an ablation over the number of circuit block layers. We focus on $K \in \{4, 5\}$ for the circuits CNN8, SU(4), and SEL-X, and test the same metrics with layers $L \in \{1, 3, 6\}$. The results provided in Table 4 show that, indeed, the discrete outputs of circuits trained with our tournament method remain more resolvable and similarly accurate even as circuit depth varies with unsurprising minor changes in overall accuracy between both methods.

Note that the improvement in performance of the tournament method over the one-hot method is even higher for shallower circuits which makes it an even more promising tool to deal with the noisy hardware currently available today, as deeper circuits allow more time for hardware noise to destroy quantum computations.

### A.1.2 HARDWARE NOISE

Our contributions compare different post-processing methods of the outputs from identically initial-ized PQCs, and are thus independent of hardware conditions—in theory. To ensure that this is this case, we ran our inference suite on both datasets with all of the circuit blocks with $K \in \{3, 4\}$ using noise models provided by IBM QisKit (Javadi-Abhari et al., 2024). This inference was performed using the same learned parameters trained under noiseless-simulation to produce the results provided in Section 4.

Table 4: Ablation showing the differing performance of circuits trained and tested with a number of 2-ring layers $L \in \{1, 3, 4, 6\}$. The main results of the paper are attained using $L = 4$. Blocks SU(4), CNN8, and SEL-X, the MNIST Digits dataset, and $K \in \{4, 5\}$ were used for the ablation.

| $K$ | Block | $L$ | Method | Resolvable-only | | | Constant | | Simulation |
|---|---|---|---|---|---|---|---|---|---|
| | | | | $A_M$ | $A_m$ | $M_r$ (↓) | $A_M$ | $A_m$ | $T$ |
| 4 | CNN8 | 1 | Tournament | 58.26 | 82.68 | 0.62 | 38.52 | 53.44 | 86.31 |
| 4 | CNN8 | 1 | One-hot | 56.04 | 80.06 | 0.39 | 23.91 | 6.73 | 83.42 |
| 4 | SU(4) | 1 | Tournament | 59.86 | 84.71 | 0.65 | 40.73 | 61.31 | 87.15 |
| 4 | SU(4) | 1 | One-hot | 57.64 | 83.38 | 0.42 | 26.44 | 10.88 | 85.75 |
| 4 | SEL-X | 1 | Tournament | 32.45 | 38.25 | 0.54 | 19.46 | 16.58 | 47.49 |
| 4 | SEL-X | 1 | One-hot | 37.63 | 42.32 | 0.27 | 13.6 | 3.73 | 50.33 |
| 4 | CNN8 | 3 | Tournament | 61.53 | 87.62 | 0.70 | 44.78 | 70.92 | 89.91 |
| 4 | CNN8 | 3 | One-hot | 62.57 | 88.97 | 0.50 | 32.76 | 22.59 | 90.4 |
| 4 | SU(4) | 3 | Tournament | 63.43 | 89.26 | 0.75 | 48.52 | 81.78 | 90.65 |
| 4 | SU(4) | 3 | One-hot | 64.96 | 89.04 | 0.54 | 36.67 | 36.68 | 90.82 |
| 4 | SEL-X | 3 | Tournament | 44.28 | 70.79 | 0.53 | 24.5 | 12.45 | 76.49 |
| 4 | SEL-X | 3 | One-hot | 39.15 | 64.28 | 0.30 | 12.46 | 0.01 | 76.67 |
| 4 | CNN8 | 4 | Tournament | 64.03 | 89.17 | 0.73 | 48.2 | 80.43 | 90.54 |
| 4 | CNN8 | 4 | One-hot | 64.38 | 89.47 | 0.54 | 36.35 | 35.86 | 90.9 |
| 4 | SU(4) | 4 | Tournament | 63.51 | 89.21 | 0.74 | 48.38 | 81.04 | 90.91 |
| 4 | SU(4) | 4 | One-hot | 64.55 | 89.86 | 0.57 | 38.07 | 44.31 | 91.48 |
| 4 | SEL-X | 4 | Tournament | 46.86 | 74.9 | 0.55 | 26.94 | 17.6 | 81.25 |
| 4 | SEL-X | 4 | One-hot | 48.04 | 75.95 | 0.37 | 19.32 | 2.59 | 84.21 |
| 4 | CNN8 | 6 | Tournament | 62.27 | 89.57 | 0.75 | 47.71 | 81.91 | 91.15 |
| 4 | CNN8 | 6 | One-hot | 62.14 | 89.62 | 0.53 | 34.18 | 28.36 | 91.34 |
| 4 | SU(4) | 6 | Tournament | 61.03 | 89.64 | 0.75 | 46.37 | 82.74 | 91.02 |
| 4 | SU(4) | 6 | One-hot | 62.12 | 89.72 | 0.53 | 34.12 | 28.04 | 91.57 |
| 4 | SEL-X | 6 | Tournament | 48.85 | 81.44 | 0.59 | 29.51 | 26.75 | 87.13 |
| 4 | SEL-X | 6 | One-hot | 50.42 | 82.67 | 0.40 | 21.02 | 1.48 | 87.96 |
| 5 | CNN8 | 1 | Tournament | 47.67 | 71.91 | 0.62 | 30.72 | 34.38 | 75.6 |
| 5 | CNN8 | 1 | One-hot | 41.57 | 63.17 | 0.33 | 14.78 | 0.21 | 69.79 |
| 5 | SU(4) | 1 | Tournament | 48.47 | 74.03 | 0.63 | 31.46 | 38.67 | 77.0 |
| 5 | SU(4) | 1 | One-hot | 43.64 | 66.19 | 0.35 | 16.44 | 0.41 | 71.76 |
| 5 | SEL-X | 1 | Tournament | 26.5 | 32.99 | 0.58 | 15.88 | 7.87 | 36.28 |
| 5 | SEL-X | 1 | One-hot | 27.91 | 30.81 | 0.21 | 7.5 | 0.5 | 32.07 |
| 5 | CNN8 | 3 | Tournament | 52.19 | 81.1 | 0.65 | 34.98 | 50.46 | 83.39 |
| 5 | CNN8 | 3 | One-hot | 51.93 | 81.48 | 0.37 | 20.46 | 0.94 | 84.81 |
| 5 | SU(4) | 3 | Tournament | 54.35 | 82.01 | 0.67 | 37.52 | 58.74 | 84.78 |
| 5 | SU(4) | 3 | One-hot | 56.11 | 84.36 | 0.40 | 23.76 | 4.0 | 86.78 |
| 5 | SEL-X | 3 | Tournament | 25.7 | 41.61 | 0.58 | 15.11 | 1.38 | 47.94 |
| 5 | SEL-X | 3 | One-hot | 25.22 | 39.07 | 0.21 | 5.55 | 0.0 | 48.38 |
| 5 | CNN8 | 4 | Tournament | 53.71 | 80.97 | 0.67 | 37.2 | 57.27 | 83.77 |
| 5 | CNN8 | 4 | One-hot | 53.68 | 83.28 | 0.39 | 22.16 | 1.97 | 85.33 |
| 5 | SU(4) | 4 | Tournament | 55.75 | 83.55 | 0.67 | 38.39 | 57.95 | 85.05 |
| 5 | SU(4) | 4 | One-hot | 56.35 | 85.33 | 0.40 | 24.13 | 4.31 | 87.78 |
| 5 | SEL-X | 4 | Tournament | 29.23 | 49.57 | 0.58 | 17.21 | 2.25 | 55.47 |
| 5 | SEL-X | 4 | One-hot | 27.61 | 45.6 | 0.22 | 6 .55 | 0.0 | 55.76 |
| 5 | CNN8 | 6 | Tournament | 53.99 | 82.15 | 0.67 | 37.39 | 58.81 | 84.53 |
| 5 | CNN8 | 6 | One-hot | 55.09 | 84.74 | 0.39 | 23.0 | 2.4 | 87.23 |
| 5 | SU(4) | 6 | Tournament | 56.66 | 84.74 | 0.70 | 40.48 | 66.49 | 86.01 |
| 5 | SU(4) | 6 | One-hot | 57.28 | 86.03 | 0.41 | 25.42 | 5.93 | 88.14 |
| 5 | SEL-X | 6 | Tournament | 30.71 | 47.26 | 0.58 | 18.29 | 7.3 | 57.62 |
| 5 | SEL-X | 6 | One-hot | 32.46 | 56.96 | 0.26 | 8.85 | 0.0 | 68.98 |

These noise models allow PennyLane to simulate the noise of real-world IBM hardware. Given the differing sizes of the circuits, we use the noise models of different IBM machines for different values of $K$—namely IBM Belem Version 2 for $K = 3$ and IBM Oslo for $K = 4$. The varying levels of noise for the varying hardware make comparing performance across different values of $K$ less productive, however the results in Table 5 still allow for a fair comparison of the primary results between the tournament and one-hot post-processing methods.

Noise-model experiments are included to illustrate relative performance trends under non-ideal conditions, not as claims of hardware readiness. Backends were selected based on Qiskit documentation at the time; updating to current backends would not affect the theoretical guarantees presented in the main text.

It can be clearly seen that hardware noise has a universally negative effect on the performance of even pretrained PQCs, however, the effectiveness of the tournament method over the one-hot method is still clearly visible. The relative performance between the two methods remains either identical or even improves for the tournament method. This is especially evident for the resolvable-only micro accuracy $R_a$, which remains better than guessing under the tournament method, but hovers near guessing level for the one-hot method. Given this improvement is attained using less samples—as evidenced by the superior resolvability ratio $M_r$—it is clear that the tournament method leads to much higher quality discrete output samples even under noisy conditions.

Table 5: Ablation on inference performance of noiseless-trained models using all six block circuit variants on simulated noisy hardware.

| Dataset | $K$ | Method | Resolvable-only | | | Constant | | Simulation |
|---|---|---|---|---|---|---|---|---|
| | | | $A_M$ | $A_m$ | $M_r$ | $A_M$ | $A_m$ | $T$ |
| Digits | 3 | Tournament | $0.37 \pm 0.04$ | $0.43 \pm 0.11$ | $0.76 \pm 0.01$ | $0.28 \pm 0.03$ | $0.38 \pm 0.10$ | $0.69 \pm 0.16$ |
| Digits | 3 | One-hot | $0.33 \pm 0.00$ | $0.33 \pm 0.01$ | $0.38 \pm 0.03$ | $0.13 \pm 0.01$ | $0.00 \pm 0.00$ | $0.67 \pm 0.20$ |
| Digits | 4 | Tournament | $0.28 \pm 0.02$ | $0.34 \pm 0.05$ | $0.53 \pm 0.02$ | $0.15 \pm 0.02$ | $0.03 \pm 0.06$ | $0.76 \pm 0.25$ |
| Digits | 4 | One-hot | $0.25 \pm 0.00$ | $0.25 \pm 0.01$ | $0.27 \pm 0.03$ | $0.08 \pm 0.01$ | $0.00 \pm 0.00$ | $0.76 \pm 0.26$ |
| Fashion | 3 | Tournament | $0.39 \pm 0.05$ | $0.50 \pm 0.11$ | $0.77 \pm 0.02$ | $0.30 \pm 0.04$ | $0.46 \pm 0.11$ | $0.70 \pm 0.16$ |
| Fashion | 3 | One-hot | $0.33 \pm 0.00$ | $0.33 \pm 0.01$ | $0.37 \pm 0.05$ | $0.13 \pm 0.02$ | $0.00 \pm 0.00$ | $0.69 \pm 0.20$ |
| Fashion | 4 | Tournament | $0.26 \pm 0.02$ | $0.29 \pm 0.05$ | $0.52 \pm 0.02$ | $0.14 \pm 0.01$ | $0.02 \pm 0.04$ | $0.71 \pm 0.24$ |
| Fashion | 4 | One-hot | $0.25 \pm 0.00$ | $0.25 \pm 0.01$ | $0.30 \pm 0.02$ | $0.08 \pm 0.01$ | $0.00 \pm 0.00$ | $0.73 \pm 0.21$ |

### A.1.3 ACTIVATION FUNCTIONS

For all four encodings, the expectation values from the PQC are activated using a sigmoid function, inspired by soft-thresholding (13) from Felsberg et al. (2009). This tempering reverses the monotonicity of the data and normalizes it, which both need to be done since the expectation value range for a quantum Pauli measurement is $[-1, 1]$, and expectation values of $-1$ and $1$ are commonly used to represent a binary $1$ and $0$, respectively (Nielsen and Chuang, 2010; Schuld and Petruccione, 2021). Activating expectation values this way enables us to reason about them as the probabilities that their qubits, when discretized through measurement, will output $1$ as opposed to $0$.

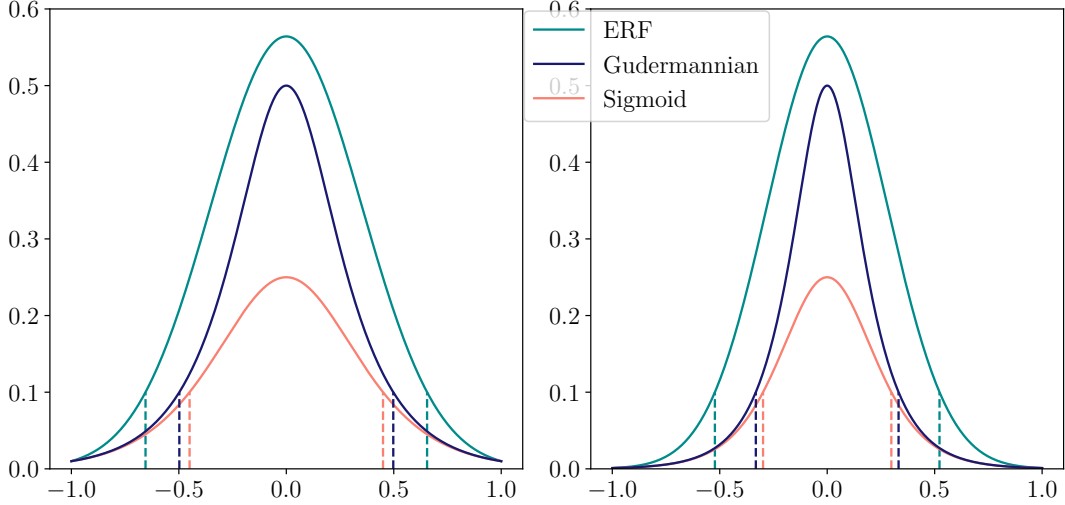

Figure 5: Plots of the derivatives of the tempering functions after being scaled such that the chosen minimal gradient occurs at -1 and 1. Dashed lines showing where each function has a gradient value of 0.1, which illustrates the relative decrease in gradient scale between the three functions. Due to scaling the functions such that they have equivalent minimum values, the graphs are equal at their endpoints, but it can be seen that the logistic and Gudermannian functions have smaller gradients which also vanish sooner than in the error function (ERF).

The secondary goal in applying such a function is to ensure that the gradients returning to the circuit are minimal near expectation values of $-1$ and $1$, and maximal near $0$, since expectations near the extrema are more likely to discretize to either $1$ or $0$, respectively, and expectation values of $0$ operate like coin-flips when discretized. Vanishing gradients from the sigmoid function have been a large enough problem in classical machine learning for them to be considered outdated (Ven and Lederer, 2021; Roodschild et al., 2020), but in this use case, it provides exactly the behavior we want. Originally, the logistic function was chosen due to the ease of calculating its gradient (Goyal et al., 2020), which, while efficient, may not lead to the optimal training behavior in quantum circuits.

There are many functions which have the required shape, with the biggest difference between them being their domains relative to their asymptotes as none reach diminished gradients in the domain $[-1, 1]$. Because of this, the inputs to the functions need to be scaled to make full use of this vanishing effect. This scaling can be such that the minimum gradient returning to the circuit is arbitrarily close to $0$, but the more this scaling is applied, the more of the input domain receives very little gradient, as shown in Fig A.1.3. In this study, we ablated over three sigmoid like functions - namely, the logistic function, the error function, and the Gudermannian function (Gambini et al., 2024) - and two minimum gradient levels for each - namely, $0.01$ and $0.001$. To calculate the scaling, we simply find the input value to the first derivative of each function that gives the minimum value we set.

Table 6: Averaged Friedman rank over all relevant statistics on CNN7 with $K = 4$ using the edge method. Highest score was chosen as the optimal.

| Function | ERF | | Linear | | Logistic | | Gudermannian | |
|---|---|---|---|---|---|---|---|---|
| Min Grad | 0.01 | 0.001 | 0.01 | 0.001 | 0.01 | 0.001 | 0.01 | 0.001 |
| F-Score↑ | **5.0** | 4.375 | 2.875 | 2.875 | 4.725 | 3.875 | 4.25 | 3.25 |

To determine which sigmoid-like function to use for the main results, we performed our ablation process on the CNN7 block from Sim et al. (2019) using $K = 4$ on the MNIST Digits dataset, shown in Table 6. We compared ERF, Gudermannian, and the logistic function at minimal gradient values of $0.01$ and $0.001$, as well as a linear monotonicity-reversing normalization.

The scores presented are Friedman-scores computed over a range metrics: both sets of micro and macro accuracies, the threshold accuracy $T$ and average distance between the top-two predictions, as well as the resolvability rate $M_r$. We use the Friedman-scores to decide on the best tempering method without focusing on a single metric.

### A.1.4 OPTIMIZATION

To ascertain the best optimization strategy before running the full experimental suite, we ran an ablation across two optimizers, four learning rate schedulers, and three learning rates. The two optimizers are standard stochastic gradient descent (SGD), invented by Robbins and Monro (1951), and Adam, invented by Diederik (2014). The learning rate schedulers we tested were an exponential scheduler, as defined by Li and Arora (2019), a cosine scheduler, as defined by Loshchilov and Hutter (2016), a piecewise scheduler, as defined by Goyal et al. (2017), and no scheduler, also called a constant scheduler.

For the exponential scheduler, there were ten total transition steps over the full six epochs, with a decay rate of $0.9$. For the cosine scheduler, the number of steps was simply the number of training steps. For the piecewise scheduler, there were three transition steps with scale factors of $0.1$ and $0.01$.

We first ran all the tests on $K = 3$ with the tournament method on the CNN7 block, shown in Table 7. To average over all the metrics, we look at the Friedman Rank (F-Rank) of each optimization strategy, which ranks the columns and averages the ranks over the rows (Friedman, 1937). Due to the tie between the piecewise scheduler and constant scheduler with the Adam optimizer, we opted to run a second set on the Adam optimizer with $K = 5$ instead. This test is shown in Table 8. As exponential decay with a learning rate of $0.01$ ranked best for $K = 5$, and nearly as well as piecewise and constant for $K = 3$, this was chosen as the optimal setup.

## A.2 Full Testing Results

Given the immense size of our full testing suite, there is no manageable way to include the full tables in this print. The results from all tests run up to this date can be found in the repo linked in Section 3.5.

## A.3 QML Block Descriptions

Here we present information about the blocks used in the 2-qubit ring structure. In this section, we will simply summarize the findings of the introductory works to justify their usage in this paper.

The first four of the blocks were found in the work by Hur et al. (2022) and showed promising results in all their experiments. In that paper, the reason each block was chosen was explained succinctly. The CNN7 and CNN8 blocks were first introduced as 4-qubit error-correcting encoders by Johnson et al. (2017). They showed the best expressibility in a study by Sim et al. (2019), leading to them being chosen by Hur et al. (2022). Expressibility, in the context of QML, is a measure of the ability of a circuit to produce a wide range of quantum states.

The SO(4) block was shown by Wei and Di (2012) to be able to implement an arbitrary SO(4) operation, and can be used to construct a fully entangled VQE. The SU(4) block was shown by Vatan and Williams (2004) and MacCormack et al. (2020) to be able to implement any arbitrary 2-qubit rotation.

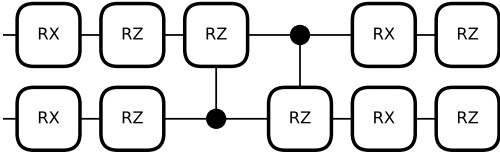

Figure 6: CNN7 Block from Sim et al. (2019), as modified by Hur et al. (2022).

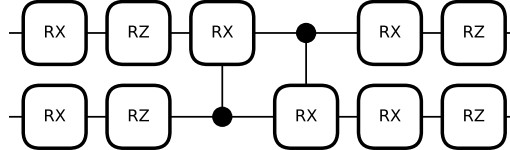

Figure 7: CNN8 Block from Sim et al. (2019), as modified by Hur et al. (2022).

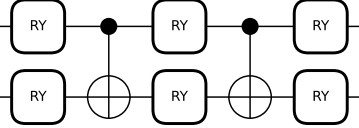

Figure 8: SO(4) Block from Wei and Di (2012), as modified by Hur et al. (2022).

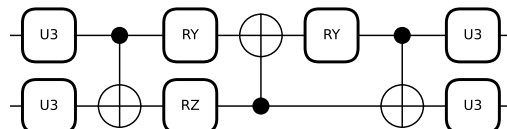

Figure 9: SU(4) Block from Vatan and Williams (2004), as modified by Hur et al. (2022).

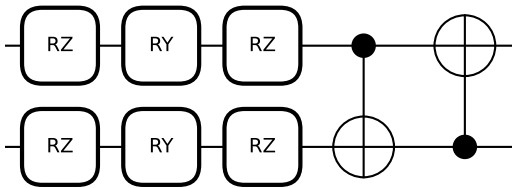

Figure 10: Strongly Entangling Layer Block with CNOT imprimitive (Sel-X) from Schuld et al. (2020).

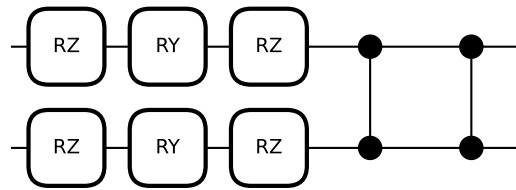

Figure 11: Strongly Entangling Layer Block with CZ imprimitive (Sel-Z) from Schuld et al. (2020).

Strongly Entangling Layers is a popular multi qubit gate-operation that is available as a callable function in the popular quantum computing package PennyLane (Bergholm et al., 2022). The setup

was invented in a paper by several of the authors responsible for the creation of PennyLane in Schuld et al. (2020), and has seen much use due to simplicity and expressibility.

Note that for the Strongly Entangling Layers block all single-qubit operations are applied before the ring of two-qubit operations rather than in alternating full block rings like in CNN7, CNN8, SO(4) and SU(4), as visualized in Figure 1. We included this block in our analysis so as to demonstrate the efficacy of the tournament encoding independent of the 2-qubit ring structure.

For even greater fairness, we include two versions using the two most common parameter-free 2-qubit operations, namely the CNOT gate and CZ gate. For more information about the gate-operations performed in these blocks, we present an accelerated introduction to quantum computing in Section A.4.

### A.4 Introduction to Quantum Machine Learning

In this section, we will give a low-level overview of the ideas from quantum computing needed to understand this work. This information is summarized from the works of Nielsen and Chuang (2010) and Schuld and Petruccione (2021) which cover it in much greater detail for the interested reader.

*Qubits* - Qubits are the quantum equivalent to a bit in classical computing. The state of a qubit is represented as a two dimensional vector in a Hilbert space, with classical states $0$ and $1$ corresponding to the quantum states $|0\rangle$ and $|1\rangle$, where

$$|0\rangle = \begin{bmatrix} 1 \\ 0 \end{bmatrix}, \text{ and } |1\rangle = \begin{bmatrix} 0 \\ 1 \end{bmatrix}. \tag{1}$$

Unlike classical bits which are binary, the state of a qubit can be any length-$1$ vector in the two-dimensional complex vector space spanned by $|0\rangle$ and $|1\rangle$.

*Gates* - Quantum gates are a quantum extension of classical reversible-logic gates.

These transform states unitarily (complex angle-preserving), so correspond to complex rotations. Simple examples include the Pauli-X, Pauli-Y, and Pauli-Z gates, written mathematically as $\sigma_1$, $\sigma_2$ and $\sigma_3$, respectively. Pauli-X, Pauli-Y, and Pauli-Z are also names for the cardinal axes within the sphere of all possible states a single qubit can take, the so-called "Bloch sphere". The matrices which represent these operations rotate a qubit $\pi$ radians around the respective axis, and all of them can be written at once as,

$$\sigma_j = \begin{pmatrix} \delta_{j3} & \delta_{j1} - i\,\delta_{j2} \\ \delta_{j1} + i\,\delta_{j2} & -\delta_{j3} \end{pmatrix}. \tag{2}$$

Many other gates exist, including gates to elicit interactions between qubits and parameterized versions of the Pauli Gates which allow rotations of arbitrary degree around their respective axes. The subset used in this work can be found in Figure 12. A brief description of these is provided, but more comprehensive details, as well as more gates, can be found in the works of Nielsen and Chuang (2010) and Schuld and Petruccione (2021).

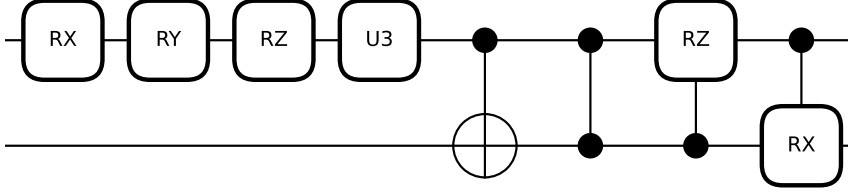

Figure 12: Subset of quantum gates used in the circuits in this paper. From left to right: Pauli-X rotation, Pauli-Y rotation, Pauli-Z rotation, 3-parameter unitary gate, CNOT, CZ, Controlled Pauli-Z rotation, and controlled Pauli-X rotation.

Of these gates, CNOT, CZ, and the controlled Pauli gates all apply their rotations conditionally based on the state of the dotted wire. In the case of the controlled Pauli rotations, this rotation is parameterized, where for CNOT and CZ, it is always a rotation of $\pi$ radians about the Pauli-X and Pauli-Z axes, respectively. The regular rotation gates are parameterized versions of their Pauli gates.

The U3 gate is a special gate which applies a parameterized Pauli-Z, followed by a parameterized Pauli-Y, and then another parameterized Pauli-Z, enabling any arbitrarily Euler rotation about the Bloch Sphere.

*Circuits* - The term "circuit" typically refers to a more complicated unitary operator built up from a number of quantum gates that are composed sequentially. The term *wire* refers to single qubits as they traverse the different operations within a circuit. The term "model" can often be interchanged with "circuit," though perhaps self-evidently, only when the model can be represented as a circuit.

*Measurements* - To extract information from a quantum circuit, a measurement of the qubits involved must be performed. A measurement has an associated Hermitian operator (real-valued eigenvalues) where the eigenvalues are the possible outcomes, and the squared length of the state projection onto one of the eigenspaces determines the probability of the corresponding outcome. Due to convention, the most common measurement in quantum computing is measurement in the "computational basis", associated with the Hermitian Pauli-Z operator (Schuld and Petruccione, 2021; Nielsen and Chuang, 2010).

A measurement always gives one of the eigenvalues of the Hermitian operator. For a Pauli-Z measurement we obtain one of two discrete outputs, +1 or -1 (mapped to the bit values 0 or 1, respectively). The *expectation value*, or the expected (average) output is then the weighted average of the outcomes. For a Pauli-Z measurement, this would be

$$E(\sigma_3) = (+1)P(+1) + (-1)P(-1) \tag{3}$$

Note that the range of this expression is $[-1, +1]$ because the eigenvalues of the $\sigma_3$ operator are $+1$ and $-1$ rather than the binary 0 and 1. If the two outcomes are equally probable, the expectation value here is $= 0$ rather than $= 1/2$, which becomes important when setting thresholds in the simulation output.

Such an expectation value can be calculated directly, though this is only possible in simulations. In an actual machine, the outputs would be the discrete values $+1$ and $-1$, so to estimate the expectation value when using a quantum computer one would need to count the outcomes and produce a point sample from a series of measurements.

Table 7: Ablation with $K = 3$ over optimizers (Opt), learning rate schedulers (LRS), and learning rates (LR). Schedulers used include Exponential Decay $exp$, Cosine Decay $cos$, Piecewise Constant $step$, Constant $reg$, and Linear Decay $lin$.

| Opt | LRS | LR | F-Rank | Valid | | | Constant | | Simulation |
|---|---|---|---|---|---|---|---|---|---|
| | | | ($\uparrow$) | $A_M$ | $A_m$ | $S$ ($\downarrow$) | $A_M$ | $A_m$ | $T$ |
| SGD | $exp$ | 0.01 | 14.875 | 54.15 | 73.73 | 116.36 | 47.02 | 72.7 | 78.33 |
| SGD | $exp$ | 0.001 | 5.375 | 50.88 | 68.84 | 125.53 | 41.6 | 63.08 | 75.02 |
| SGD | $exp$ | 0.0001 | 2.0 | 42.04 | 50.62 | 129.33 | 33.5 | 44.78 | 57.68 |
| SGD | $reg$ | 0.01 | 17.0 | 54.16 | 73.77 | 116.32 | 47.04 | 72.77 | 78.39 |
| SGD | $reg$ | 0.001 | 7.625 | 50.92 | 68.99 | 125.43 | 41.66 | 63.27 | 74.84 |
| SGD | $reg$ | 0.0001 | 3.125 | 42.1 | 50.76 | 129.37 | 33.53 | 44.85 | 58.29 |
| SGD | $step$ | 0.01 | 17.0 | 54.16 | 73.77 | 116.32 | 47.04 | 72.77 | 78.39 |
| SGD | $step$ | 0.001 | 7.625 | 50.92 | 68.99 | 125.43 | 41.66 | 63.27 | 74.84 |
| SGD | $step$ | 0.0001 | 3.125 | 42.1 | 50.76 | 129.37 | 33.53 | 44.85 | 58.29 |
| SGD | $cos$ | 0.01 | 16.5 | 54.16 | 73.77 | 116.32 | 47.04 | 72.77 | **78.4** |
| SGD | $cos$ | 0.001 | 6.875 | 50.92 | 68.98 | 125.43 | 41.66 | 63.23 | 74.85 |
| SGD | $cos$ | 0.0001 | 2.375 | 42.09 | 50.74 | 129.36 | 33.53 | 44.84 | 58.28 |
| SGD | $lin$ | 0.01 | 15.625 | 54.15 | 73.75 | 116.35 | 47.02 | 72.71 | 78.35 |
| SGD | $lin$ | 0.001 | 6.125 | 50.89 | 68.84 | 125.52 | 41.61 | 63.11 | 75.03 |
| SGD | $lin$ | 0.0001 | 2.5 | 42.05 | 50.65 | 129.34 | 33.5 | 44.79 | 57.72 |
| Adam | $exp$ | 0.01 | 16.75 | 54.62 | 69.29 | **114.15** | 48.49 | 68.54 | 76.66 |
| Adam | $exp$ | 0.001 | 19.875 | **55.49** | **75.16** | 114.41 | 48.92 | 74.25 | 78.26 |
| Adam | $exp$ | 0.0001 | 9.5 | 51.7 | 70.33 | 123.64 | 42.71 | 65.42 | 75.14 |
| Adam | $reg$ | 0.01 | 17.5 | 54.65 | 69.3 | 114.19 | 48.49 | 68.63 | 76.64 |
| Adam | $reg$ | 0.001 | **21.125** | **55.49** | 75.07 | 114.35 | **48.94** | **74.29** | 78.27 |
| Adam | $reg$ | 0.0001 | 12.125 | 51.76 | 70.52 | 123.53 | 42.78 | 65.6 | 75.17 |
| Adam | $step$ | 0.01 | 17.5 | 54.65 | 69.3 | 114.19 | 48.49 | 68.63 | 76.64 |
| Adam | $step$ | 0.001 | **21.125** | **55.49** | 75.07 | 114.35 | **48.94** | **74.29** | 78.27 |
| Adam | $step$ | 0.0001 | 12.125 | 51.76 | 70.52 | 123.53 | 42.78 | 65.6 | 75.17 |
| Adam | $cos$ | 0.01 | 17.375 | 54.65 | 69.29 | 114.19 | 48.49 | 68.64 | 76.65 |
| Adam | $cos$ | 0.001 | 21.0 | **55.49** | 75.08 | 114.35 | **48.94** | **74.29** | 78.27 |
| Adam | $cos$ | 0.0001 | 11.625 | 51.76 | 70.52 | 123.53 | 42.78 | 65.61 | 75.16 |
| Adam | $lin$ | 0.01 | 17.75 | 54.63 | 69.3 | **114.15** | 48.5 | 68.59 | 76.66 |
| Adam | $lin$ | 0.001 | 20.5 | **55.49** | **75.16** | 114.41 | 48.93 | 74.27 | 78.26 |
| Adam | $lin$ | 0.0001 | 10.375 | 51.71 | 70.37 | 123.63 | 42.72 | 65.45 | 75.14 |

Table 8: Ablation with $K = 5$ using Adam over learning rates (LR), learning rate schedulers (LRS). Schedulers used include Exponential Decay $exp$, Cosine Decay $cos$, Piecewise Constant $step$, Constant $reg$, and Linear Decay $lin$.

| Opt | LRS | LR | F-Rank | Valid | | | Constant | | Simulation |
|---|---|---|---|---|---|---|---|---|---|
| | | | ($\uparrow$) | $A_M$ | $A_m$ | $S$ ($\downarrow$) | $A_M$ | $A_m$ | $T$ |
| Adam | $exp$ | 0.01 | **11.0** | **49.25** | 80.43 | 151.77 | **33.16** | 47.5 | **83.99** |
| Adam | $exp$ | 0.001 | 5.125 | 47.62 | 78.95 | 155.31 | 31.41 | 40.19 | 82.9 |
| Adam | $exp$ | 0.0001 | 2.375 | 36.63 | 65.06 | 166.9 | 22.41 | 13.36 | 72.54 |
| Adam | $cos$ | 0.01 | 9.625 | 49.23 | 80.36 | 151.79 | 33.14 | 47.49 | 83.92 |
| Adam | $cos$ | 0.001 | 6.875 | 47.62 | 78.92 | 155.21 | 31.43 | 40.14 | 82.99 |
| Adam | $cos$ | 0.0001 | 3.75 | 36.75 | 64.85 | 166.84 | 22.48 | 13.67 | 72.7 |
| Adam | $step$ | 0.01 | 9.625 | 49.23 | 80.36 | 151.79 | 33.14 | 47.49 | 83.92 |
| Adam | $step$ | 0.001 | 6.875 | 47.62 | 78.92 | 155.21 | 31.43 | 40.14 | 82.99 |
| Adam | $step$ | 0.0001 | 3.75 | 36.75 | 64.85 | 166.84 | 22.48 | 13.67 | 72.7 |
| Adam | $reg$ | 0.01 | 10.625 | 49.24 | **80.44** | 151.78 | 33.15 | **47.54** | 83.94 |
| Adam | $reg$ | 0.001 | 6.75 | 47.62 | 78.96 | 155.22 | 31.42 | 40.33 | 82.98 |
| Adam | $reg$ | 0.0001 | 3.5 | 36.75 | 65.11 | 166.87 | 22.48 | 13.61 | 72.71 |
| Adam | $lin$ | 0.01 | 10.25 | 49.23 | 80.36 | **151.76** | **33.16** | 47.46 | 83.97 |
| Adam | $lin$ | 0.001 | 5.25 | 47.6 | 78.78 | 155.27 | 31.41 | 40.32 | 82.9 |
| Adam | $lin$ | 0.0001 | 2.875 | 36.65 | 64.97 | 166.9 | 22.42 | 13.44 | 72.57 |