# OpenReview forum: "Quan-dorcet: Tournament-Based One-vs-One Quantum Classification for Robust Single-Shot Inference"
_ICLR.cc/2026/Conference — Submitted to ICLR 2026_

### Official Review · Reviewer_AhDi · 2025-11-03

**Soundness:** 3
**Presentation:** 3
**Contribution:** 2
**Rating:** 6
**Confidence:** 4

**Summary:**

The paper introduces a novel output encoding technique for QML in multi-class classification tasks. Specifically, the authors propose a decision aggregation framework based on round-robin tournament scoring, offering an alternative to traditional encoding schemes such as Gray and binary encoding. A key feature of the proposed approach is its reliance on one-shot encoding, where a single measurement is used to determine the class label, rather than computing expectations over multiple shots. This design aligns well with practical constraints in QML, contributing to more efficient inference.
In addition to the theoretical framework, the paper presents empirical evaluations across multiple datasets, demonstrating that the proposed method consistently outperforms existing encoding strategies.

**Strengths:**

Reducing the sample complexity or shot complexity of QML for multiclass problems is an interesting and important problem that this paper addresses. The numerical experiments clearly demonstrate that the proposed method outperforms existing approaches for output encoding. Overall, the paper is relatively well written and easy to follow

**Weaknesses:**

The paper focuses on a single-shot approach, which is nice. But I wonder if a comparison with a few-shot measurements would make sense and give improvements. Essentially, a comprehensive discussion on the number of measurement shots for output encoding seems lacking in this paper.

**Questions:**

Can you provide any theoretical justification comparing the single shot with the fixed multi-shot approach?

---

> ### Author Response · Authors · 2025-11-21
>
> Thank you for highlighting the novelty and practical alignment of our approach. We appreciate your recognition that the tournament-based framework addresses real constraints in QML by improving single-shot inference efficiency, and your acknowledgment of the empirical validation across multiple datasets.
> Below, we respond to your suggestions regarding shot count analysis and theoretical justification.
>
> **Discussion on Shot Count:** We appreciate this suggestion. Our evaluation of single-shot performance is intrinsically linked to few-shot behavior: the reported metrics (e.g., shot accuracy and resolvability ratio) are computed over 100 shots per input sample, which allows us to analyze both single-shot quality and its correlation with multi-shot inference. In general, accuracy improves monotonically with additional shots, but at the cost of increased measurement overhead. Our focus on single-shot inference reflects a worst-case baseline—if a method performs well with one shot, it will only improve with more. We will make this rationale clearer in the revised manuscript.
>
> **Theoretical Justification:** The reasoning follows from the above: more shots always improve accuracy, but at higher cost. By improving single-shot reliability, our method reduces the number of shots needed to achieve a given confidence level, minimizing wasted compute. This makes single-shot a meaningful theoretical benchmark for comparing output encodings under strict sampling constraints.

---

### Official Review · Reviewer_BZ68 · 2025-11-04

**Soundness:** 2
**Presentation:** 3
**Contribution:** 2
**Rating:** 4
**Confidence:** 4

**Summary:**

This paper presents a new decision aggregation framework, QUAN-DORCET to address the critical sampling bottleneck and fragile output encodings in multiclass quantum machine learning under limited measurement budgets, especially for single-shot inference. The main contribution is the replacement of traditional global output schemes with a round-robin tournament structure where each output qubit performs a binary comparison between a pair of classes. The final prediction is a Condorcet-style winner determined by majority wins across all pairwise comparisons which theoretically converges to full resolvability as the number of classes increases. The authors also develop a unique, differentiable training method that embeds the pairwise comparisons into a continuous simplex using a symmetric cross-entropy loss and empirically demonstrate that this approach significantly improves both shot resolvability and accuracy compared to baselines in few-shot regimes.

**Strengths:**

The paper introducing a novel output encoding for VQC based on external political/tournament theory (Condorcet's criterion) to solve the intrinsic quantum problem of low single-shot resolvability. The core innovation is leveraging the statistical robustness of the tournament structure, which, unlike one-hot encoding does not suffer from exponentially vanishing resolvability and, unlike binary encoding is less susceptible to single-bit noise causing large semantic misclassifications. The quality of the work is substantiated by the comprehensive theoretical analysis (Section 3.2) and the technical development of a non-trivial, end-to-end differentiable training procedure that successfully maps the $K(K-1)/2$ binary outputs to a continuous simplex for gradient-based optimization. The writing is clear and provides sound motivation explicitly highlighting the trade-offs of all incumbent encoding methods in Table 1.

**Weaknesses:**

The most significant weakness is the fundamental issue of quadratic scaling in the required number of qubits as $K$ classes require $K(K-1)/2$ output qubits meaning the approach is constrained to small class counts (e.g., $K\le6$ in the experiments) and cannot scale to large classification problems on near-term hardware. While the method aims for robustness on real-world devices, all model training and performance evaluations are conducted exclusively under noiseless simulation. Also, a critical and missing piece of the empirical quality is a systematic study of how the Condorcet-style aggregation handles realistic hardware noise (e.g., bit-flip or depolarizing channels) in comparison to the binary/Gray codes it claims to outperform in terms of noise robustness. Overall, the paper should address the potential for Condorcet cycles (tournament paradox) in the measured results as the existence and frequency of these cycles would determine the fundamental practical limit of the method's resolvability under non-ideal (noisy) conditions.

**Questions:**

A key question for the authors concerns the empirical analysis of the Condorcet paradox,

a) Could the authors provide data on the frequency of non-unique winners (cycles or ties) in the tournament aggregation? As this is a vital component of the non-resolvable shots and have they considered using a tie-breaking rule or a ranking method (like the Schulze method) to maximize the practical resolvability?

b) Given that the robustness to bit-level noise is a central claim against binary encodings, the authors should perform an actionable study by including results for all encodings under a simulated hardware noise model (e.g., $1\%$ depolarizing noise on the output measurement qubits) to confirm the robustness advantage in the setting where it is most needed. As a suggestion to overcome the $\mathcal{O}(K^2)$ qubit requirement, have the authors explored an alternative sparse tournament structure, such as a hierarchical elimination or a Tournament-of-Champions (ToC) scheme, and can they provide an analysis on the trade-off between the decreased resource cost and the expected decrease in single-shot resolvability?

---

> ### Author Response · Authors · 2025-11-21
>
> Thank you for your detailed and insightful review. We greatly appreciate your recognition of the core innovation—leveraging tournament theory to overcome single-shot resolvability challenges—and your acknowledgment of the differentiable training procedure as a non-trivial technical contribution. Your comments on the clarity of our writing and the explicit framing of trade-offs validate the effort we invested in making this work both rigorous and transparent. Below, we address your points on scaling, hardware applicability, and Condorcet cycles.
>
>
> **Scaling and Resource Trade-offs, Hardware Applicability, and Noise Robustness:** As many reviewers raised similar concerns, we have addressed these common through-lines in our general comment, which we hope provides comprehensive answers to your primary points.
> In response to your question on bit-level noise, we have distinguished more clearly in many statements throughout the paper that the bit-level noise we address comes from quantum sampling variability and not as a byproduct of hardware limitations.
>
> **Condorcet Cycles and Tournament Structure:** We appreciate this observation. Our method references Condorcet theory for its convergence properties under random tournaments, but it does not require a Condorcet winner to produce a prediction. We use a maximum-votes aggregation rule (Copeland-style), which guarantees a prediction even when cycles occur. For clarity, we have revised the manuscript to state explicitly that cycles do not affect inference in our framework in Section 3.2.
>
> These guarantees produced what we believe to be publishable results on their own, and we view this work as a necessary foundation for future research. We are actively exploring extensions such as cycle-aware heuristics, the Schulze method, and other ideas from Moon’s _Topics on Tournaments_ (1968). We also appreciate the suggestion to investigate hierarchical or sparse tournament structures, which could reduce resource cost while maintaining resolvability. These, along with several other directions suggested by reviewers, are highlighted in the revised manuscript as promising avenues for future work.
>
> For your question about the frequency of ties, we point out that this is simply $1-R$.

---

> > ### Comment · Reviewer_BZ68 · 2025-11-27
> > **Response by Reviewer BZ68**
> >
> > Thank you for the additional clarifications. I appreciate the effort the authors put into addressing the concerns raised. However, after reviewing the responses, I am not fully convinced that my main concerns have been resolved. I will therefore maintain my original score.

---

### Official Review · Reviewer_JMzA · 2025-11-05

**Soundness:** 3
**Presentation:** 1
**Contribution:** 2
**Rating:** 4
**Confidence:** 4

**Summary:**

The papr introduces "Quan-dorcet" a round-robin tournament aggregation technique for multiclass quantum classification. Quan-dorcet targets the fragility of existing output encodings in QML, specifically under single.few-shot measurement constraints. In the following I provide the strengths and weaknesses of the paper.

**Strengths:**

1. **Addressing an important bottleneck:** Authors address an important bottleneck in current QML architectures which is poor reliability and the requirement of high shot.

2. **The encoding** The tournament is well motivated. Going beyond ad hoc output mappings and leveraging robust statistical voting mechanisms.

3. **Comparison** Empirical comparisons against standard output encodings (one-hot, binary, Gray) on quantum circuits for MNIST digits/fashion datasets are extensive, covering multiple circuit types and parameter regimes.

4. **Code availability** The authors have publicly released their code, demonstrating a commitment to open-source practices and enabling reproducibility.

**Weaknesses:**

## **Quadratic resource scaling:**
The method requires a number of quantum wires/qubits that scales quadratically with the number of classes: for K classes ~K(K-1)/2, pairwise comparisons are encoded. This makes the approach impractical for even moderate sized output spaces and restricts applicability to small problems. The text neither addresses this limitation nor empirical studies on the maximal class count achievable on actual hardware.

## **Lacking QPU execution:**
All the main experiments are performed under noiseless simulation; only limited inference ablations are reported using noise models from IBM Qiskit. There is no demonstration of end-to-end training or inference on real quantum processors. The authors should provide a detail characterization of the Quan-dorce's robustness under device noise and decoherence effects.

## **Scalability:**
The computational cost is high around "100 kCPU-hours". As stated by the authors, the approach necessitated "unforeseen compute limitations" that prevented reporting results for all circuit blocks and larger K in time for the submission. I believe this demonstrates not only scalability barriers for NISQ devices but also for classical simulation pipelines.

## **Lacking analysis of class imbalance and semantic overlap:**
The framework does not explore:
- Class imbalance, where some classes are much less represented than others, can severely impair prediction accuracy because minority classes may rarely win pairwise matchup. I believe this can lead to majority-win bias and poor generalization.
- Semantic overlap, where different classes share similar characteristics, can lead to  ambiguous or non-separable boundaries.

## **Clarity:**
The manuscript is technically sound but sometimes impenetrable to non-specialists.  Some claims about tournament theory and Condorcet aggregation would benefit from clearer intuitive explanations and more concrete worked examples.

## **Lack of references**
In introduction:
- The second sentence does not provide any reference. Such as the claim that PQC used for encoding input data.
- The term "tunable gate operations" is vague.
- The sentence `practical implementations face a significant challenge in the form of a sampling bottleneck` neither provide any justification why this bottleneck appears nor it provide any relevant information.
- The claim `the proportion of resolvable outputs.. vanishes exponentially with the number of classes, making inference increasingly unreliable` is made without any references or explanation. I encourage the author should provide more references and explanations in the introduction.

**Questions:**

As noted in the weaknesses above, I would like to pose the following questions and suggestions to the authors:

## **Resource Scaling:**
   - Can you propose, analyze, or empirically test methods to reduce the quadratic scaling of qubit requirements?
- For example, could some pairwise decisions be encoded or aggregated classically, or can hybrid output encodings balance accuracy with qubit economy?
- What is the largest class count (K) for which the method remains practical on current or near-term hardware?

## **Quantum hardware execution:**
   - Do you have plans to implement the tournament method on real quantum processors, and if so, what are the expected resource bottlenecks and noise impacts?
- Could you provide results, or at least simulated characterizations, for your method under realistic device noise and decoherence, especially regarding accuracy and resolvability?

## **Scalability:**
   - Can you clarify the extent of computational resources required across architectures, and suggest optimizations to the classical simulation pipeline?
- What are the directions for scaling up your method on either quantum or classical backends?

## **Class Imbalance and semantic overlap:**
   - How does your framework handle class imbalance, where some classes are underrepresented and risk being overlooked in majority voting?
- Could you design experiments with imbalanced or overlapping class distributions, and report rates of ties and ambiguous predictions (such as `Condorcet cycles`)?

## **Clarity:**
   Can you expand the descriptions of tournament theory and Condorcet mechanisms with visual example (e.g., for a small 3-class case)?

## **References:**
   Can you add more references to support the foundational claims in the introduction, especially regarding PQC input encoding, sampling bottlenecks, and exponential validity decay with the number of classes?

---

> ### Author Response · Authors · 2025-11-21
>
> Thank you for recognizing the core strengths of our work. We deeply appreciate your emphasis on the importance of addressing the sampling bottleneck in QML and your acknowledgment that our tournament-based encoding moves beyond ad hoc mappings toward a principled, statistically grounded approach. Your note on the breadth of our empirical comparisons and our commitment to open-source reproducibility reflects a clear understanding of what we aimed to achieve.
> Your main concerns about quadratic scaling and applicability on modern hardware are common through-lines in the reviews, so we have chosen to address them in our general statement for conciseness. Below, we respond to your specific questions:
>
> **Scaling**
>     _Can you propose, analyze, or empirically..._ This is an important direction for future work. In response to multiple requests for clarification, we have noted this explicitly in the revised manuscript.
>     _For example, could some pairwise decisions be encoded or aggregated ..._ Our post-processing is entirely classical aside from the PQC blocks, which remain constant across all embeddings. Hybrid approaches represent a separate research question rather than a necessary extension of our contribution.
>     _What is the largest class count ($K$)..._ In principle, any quantum computer with more than $K(K-1)/2$ qubits could implement the method for $K$ classes, subject to hardware noise and fidelity constraints.
>
> **QPU Execution**
>     _Do you have plans to implement the tournament method on real quantum processors, and if so, what are the expected resource bottlenecks and noise impacts?_ We do not have immediate plans to run on quantum hardware. As noted in the general comment, the circuits used in our manuscript are intentionally not unique to our contribution. Bottlenecks would therefore mirror those typically encountered in NISQ PQC training, which are well documented and outside the scope of this work.
>
>    _Could you provide results, or at least simulated characterizations, for your method under realistic device noise and decoherence, especially regarding accuracy and resolvability?_ Our method builds on existing PQC architectures and is supported by Malinovsky \& Moon's theoretical guarantees. Appendix A.1.2 includes inference under IBM Qiskit noise models, which reflects what could be expected on current hardware. We do not recommend using current quantum hardware for machine learning tasks at this stage.
>
> **Compute Cost**
>     _Can you clarify the extent of computational resources required across architectures, and suggest optimizations to the classical simulation pipeline?_ The reported 100 kCPU-hours reflect the breadth of our experimental suite: 1,260 models trained on 32-core processors, averaging $\approx 2.5$ hours per model. This cost is dominated by the number of models and the inherent difficulty of quantum simulation, not any inefficiency introduced by our method. All experiments were run in JAX, which is highly optimized for quantum operations and backpropagation.
>     _What are the directions for scaling up your method on either quantum or classical backends?_ Our post-processing method is computationally inexpensive—comparable to a classical loss function. The high experimental cost arises from simulation complexity and the volume of models trained. Improving the state of the art in quantum simulation or hardware performance lies outside the scope of this work.
>
> **Class Imbalance and Semantic Overlap**
>     _How does your framework handle class imbalance, where some classes are underrepresented and risk being overlooked in majority voting?_ Please see our general comment on Realism, Imbalance, and Overlap.
>     _Could you design experiments with imbalanced or overlapping class distributions, and report rates of ties and ambiguous predictions (such as Condorcet cycles)?_ Imbalanced data is a reasonable next step but resides outside the scope of this contribution. Ambiguous predictions correspond to the complement of the resolvability ratio, which is theoretically bounded. Future work may explore cycles as a training heuristic or as a tighter bound on shot requirements.
>
> **Other**
>     _Can you expand the descriptions of tournament theory..._ We appreciate this suggestion. Figure~1 was intended to illustrate this concept. We have also expanded Section 3.2 to increase understanding of these concepts. If these do not suffice, we welcome clarification on which aspects require expansion.
>     _Can you add more references to support the foundational..._ We will duplicate and expand relevant references in the introduction for clarity.
>     Specifically, your initial review asked for several clarifications in the opening paragraph which, upon a reread, made sense. It needed to be clearer, so we have modified this first paragraph to ensure no false claims or misunderstandings. Please let us know if further modifications are needed.

---

> > ### Comment · Reviewer_JMzA · 2025-11-25
> >
> > Thank you again for the rebuttal. The clarifications and proposed added material might improve the paper, but several of my core concerns remain only partially addressed. In the following I provide my comment and some new issues arise from the specifics of the hardware discussion, which I missed last time.
> >
> > ## Quantum hardware execution
> >
> > The response by the authors indicates, there are no immediate plans to run the method on QPU and iterates that the current NISQ devices are not recommended for QML, but this mostly restates the wellknown limitation than discussing the specific implication of the work. Such as:
> >
> > 1. The only QPU related evidence (in A.1.2) is via noise models based on *ibm_oslo* and *ibm_belem*, both of which are deprecated ((kindly check this source: https://quantum.cloud.ibm.com/docs/en/guides/retired-qpus)). As a result, the reported noise-model experiments in the appendix provides an incomplete characterization that does not reflect the noise profiles of current backends.
> >
> > 2. The statement “...we do not recommend using current quantum hardware for machine learning tasks at this stage” is somewhat out of alignment with the broader QML literature such as (Havlíček, Vojtěch, et al. "Supervised learning with quantum-enhanced feature spaces." Nature 567.7747 (2019): 209-212.) and (Farhi, Edward, and Hartmut Neven. "Classification with quantum neural networks on near term processors." arXiv preprint arXiv:1802.06002 (2018).) to mention a few, where they have already demonstrated nontrivial classification tasks on real devices.
> >
> > In summary, my original concern was not whether full-scale training is feasible on current hardware, but whether you can provide any concrete evidence or some analysis of how the presented scheme behaves under realistic noise beyond these limited and deprecated-noise-model experiments. I was specifically expecting​ **a more targeted noise studies e.g. the scaling of resolvability and accuracy with depth, width, shot count, and $K$ under representative  currently available backends.**
> >
> > ## Scalability
> > At present, in the rebuttal you only responded with: simulation is expensive, hardware is not ready, and scaling is future work. This significantly limits the practical impact. The readers are left without clear guidance on the regimes (ranges of $K$, circuit depth and number of gates, dataset characteristics) where this method is realistically usable, even in simulation, or how to mitigate the quadratic scaling in practice.
> >
> > ## Class imbalance and semantic overlap
> > The statement that extensions such as weighted voting or custom tie-breaking rules are “future work” does not provide any practical guidance on when the current method is likely to behave robustly versus when it may systematically fail. Even without new experiments, it would strengthen the paper to (1) discuss qualitatively how the resolvability ratio and majority outcome are expected to behave under imbalance and overlap. (2) Clearly flag in the main text that the present evaluation is restricted to balanced, relatively clean-label settings and that robustness to imbalance and overlapping classes is an open question.

---

> > > ### Author Response · Authors · 2025-11-26
> > >
> > > Thank you for your continued engagement and for clarifying your concerns. We appreciate your emphasis on practical applicability. Our intent, however, is to establish a theoretical foundation for encoding strategies under strict sampling constraints—a prerequisite for future hardware studies. Would you consider that clearly stating these boundaries and future directions addresses your concern?
> > > Below, we address your points concisely and outline the changes we will make:
> > >
> > > **1. Quantum Hardware Execution**
> > >
> > > Our guarantees—such as resolvability converging to unity as $K\to\infty$ (Malinovsky \& Moon, 2024)—are hardware-independent and noise-independent, derived entirely from tournament theory. These results hold regardless of backend fidelity and are not tied to accuracy, as illustrated in Figure 3 (Left), where accuracy and resolvability vary across all four encodings.
> > >
> > > The noise-model experiments in Appendix A.1.2 were included only to show relative trends, not readiness for deployment. We will clarify this explicitly and note that hardware validation is future work.
> > >
> > > Change: Add statements in Section 5 and Appendix A.1.2 explaining the rationale for using noise models and why theoretical guarantees do not depend on backend fidelity.
> > >
> > > We can update to more recent noise models if strongly requested, but these simulations are slow due to JAX’s limited support for noise modeling compared to Qiskit. Our apologies for referencing deprecated backends; we relied on Qiskit’s documentation at the time.
> > >
> > > **2. Scalability**
> > >
> > > Quadratic scaling is a structural trade-off against exponential sampling cost—a central point of the paper.
> > >
> > > Change: Add a short subsection in Section 5 outlining:
> > >
> > > * Practical simulation regimes (e.g., small $K$, shallow circuits).
> > > * Emphasize that our method targets low-shot inference, which remains relevant even in simulation.
> > >
> > >
> > > **3. Class Imbalance \& Semantic Overlap**
> > >
> > > While new experiments are beyond scope, we will:
> > >
> > > * Explicitly flag in the main text that evaluation assumes balanced datasets with clean labels.
> > > * Add a qualitative discussion of expected behavior:
> > >
> > >    * Imbalance may bias majority voting toward dominant classes.
> > >    * Overlap increases cycle frequency, which our framework currently discards as “unresolvable.”
> > >
> > >
> > >
> > >
> > > **Assertive Clarification**
> > >
> > > Although contemporary QML literature often blends algorithm design with hardware engineering (e.g., error correction, noise resilience), these fields can—and should—be studied independently. Foundational algorithms such as Shor’s and Grover’s were celebrated despite being impractical on early hardware because they offered provably beneficial future directions. Our work aims to do the same for multiclass quantum classification under strict sampling constraints. We believe this distinction is critical for advancing theory without conflating it with short-term hardware limitations.
> > >
> > > **Closing Note**
> > >
> > > Our work is not intended as an immediate hardware-ready solution but as a conceptual baseline for future QML research. We believe these clarifications and changes will make the scope and limitations transparent while preserving the theoretical contribution.

---

### Official Review · Reviewer_XFm6 · 2025-11-06

**Soundness:** 2
**Presentation:** 2
**Contribution:** 2
**Rating:** 2
**Confidence:** 3

**Summary:**

This paper proposes a novel output encoding framework, "Quan-Dorcet," for quantum multiclass classification. The method is based on pairwise round-robin tournament comparisons, aiming to improve the accuracy and "resolvability" of single-shot and few-shot inference. The authors introduce "shot resolvability" as a key metric and demonstrate through simulation that their method outperforms one-hot and binary encodings for a small number of classes (K≤6).

**Strengths:**

The paper addresses the challenge of multiclass classification in quantum machine learning (QML), particularly the issues of fragile output encodings and high sampling demands under few-shot or single-shot inference. The authors propose a tournament-based one-vs-one encoding scheme wherein each qubit corresponds to a binary comparison between a pair of classes; class prediction is obtained via a round-robin vote (Condorcet winner) across all pairwise outcomes. This decouples the multiclass output into many simpler binary decisions embedded within a shared entangled quantum state, which the authors argue improves “shot resolvability” (probability a single measurement yields a valid class) and accuracy under measurement-limited settings. Empirical results demonstrate that the scheme outperforms standard one-hot or binary encoding architectures in few-shot/single-shot regimes, suggesting this paradigm as a robust path for quantum classifiers with fewer measurements.

**Weaknesses:**

Scalability Problem: This is the most critical issue. As the authors admit in Section 5 ("Limitations"), the required number of qubits (wires) $W$ scales quadratically with the number of classes $K$ ($W = \binom{K}{2} = K(K-1)/2$).

Impact on Practicality: This makes the method practically infeasible. While binary encoding requires only $\lceil \log_2 K \rceil$ qubits, the proposed method requires 45 qubits for $K=10$ and nearly 5,000 qubits for $K=100$. This is unattainable on near-term (or even mid-term) quantum hardware.

Poor Trade-off: The paper claims to solve the "sampling bottleneck" but introduces a far more severe "qubit resource bottleneck." This trade-off (exchanging sampling efficiency for an exponentially increasing qubit requirement) is unacceptable in practice.

Incomplete Experiments: As noted in the footnotes of Table 2 and Table 3, the authors admit that "Due to unforeseen compute limitations," the results for $K=6$ are incomplete, tested on only one circuit, and that the rest "will be ready by rebuttal period." This indicates the submission is incomplete work.

**Questions:**

While this contribution is creative and potentially impactful, several limitations and open concerns remain:

1. Limited demonstration of practical quantum advantage

The experiments, while promising, appear to be simulation-based (no demonstrated real quantum hardware results). Thus, real-world factors (noise, decoherence, measurement error, circuit overhead) are not fully addressed.

The improvement in “shot resolvability” is compelling, but it remains unclear how that metric translates into end-to-end system performance, especially when scaling to larger class sets or higher dimensional inputs.

2. Scalability issues not thoroughly addressed

For  𝐾 classes, one-vs-one induces  K*(K-1)/2 binary comparisons/qubits (or circuits). The paper should more clearly analyse the resource scaling (qubits, gates, measurement overhead) for large 𝐾, and whether the tournament cost outweighs encoding benefits.

The assumption that a unique Condorcet winner will emerge reliably may break down in practice under noisy or ambiguous class boundaries; the authors should discuss scenarios where majority voting may fail or require tie-break strategies.

3. Baseline comparisons and alternative encodings

Although one-hot and binary encodings are compared, more recent and sophisticated quantum multiclass classifier encoding schemes (e.g., amplitude encoding, mixed‐state discriminators) are not deeply benchmarked. Without comparison to strong state-of-the-art quantum multiclass methods, the improvement claim is less convincing.

Moreover, many classical multiclass frameworks (ensemble binary classifiers, one-vs-one classical SVMs) employ similar architectural breakdowns; a comparison of quantum vs classical one-vs-one paradigms would strengthen significance.

4. Theoretical justification of single-shot improvements

The notion of “shot resolvability” is interesting but currently heuristic. A deeper theoretical analysis of how measuring fewer shots yields reliable class output (given quantum measurement statistics, error rates) would improve confidence.

Also, the impact of entanglement and shared statewide encoding on error propagation among the binary comparators is not fully addressed.

5. Application and realism of datasets/inputs

The datasets used, while unspecified here in detail, likely involve small-scale toy problems under simulation. It remains unclear how the method performs on large-input real-world classification tasks (e.g., image datasets with many classes and high dimensionality).

The authors should discuss how measurement budget, class imbalance, noise, and decoherence would influence performance in near-term quantum devices.

**Relevant References for Inclusion**

To strengthen the literature context and demonstrate awareness of related work, the authors should consider citing the following:

Du, Y., Yang, Y., Hsieh, M.-H., & Tao, D. (2023). Problem-Dependent Power of Quantum Neural Networks on Multi-Class Classification. Phys. Rev. Lett. 131, 140601.

Bokhan, D., Mastiukova, A. S., Boev, A. S., Trubnikov, D. N., & Fedorov, A. K. (2022). Multiclass classification using quantum convolutional neural networks with hybrid quantum-classical learning. arXiv:2203.15368.

Useche, D. H., Quiroga-Sandoval, S., Molina, S. L., Vargas-Calderón, V., Ardila-García, J. E., & González, F. A. (2025). Quantum generative classification with mixed states. arXiv:2502.19970.

Delilbasic, A., Le Saux, B., Riedel, M., Michielsen, K., & Cavallaro, G. (2023). A Single-Step Multiclass SVM based on Quantum Annealing for Remote Sensing Data Classification. arXiv:2303.11705.

Cruzeiro, E. Z., De Mol, C., Massar, S., & Pironio, S. (2023). Quantum-inspired classification based on quantum state discrimination. arXiv:2303.15353.

Riaz, F., Abdulla, S., Suzuki, H., Ganguly, S., Deo, R. C., & Hopkins, S. (2023). Accurate Image Multi-Class Classification Neural Network Approaches including quantum variations. Sensors, 23(5):2753.

---

> ### Author Response · Authors · 2025-11-21
>
> Thank you for your clear and accurate summary of our work. We greatly appreciate that you recognized the core motivation—addressing fragile encodings and sampling bottlenecks in multiclass QML—and captured how our tournament-based approach improves single-shot reliability without sacrificing global entanglement. Your framing of shot resolvability and the practical implications of reducing measurement overhead reflects a deep understanding of what we aim to contribute.
> Below, we respond to your specific concerns.
>
> **Scalability Problem:** Please see our general comment on the scalability trade-off.
>
> _...more clearly analyse the resource scaling...:_ It is not possible to definitively state whether exponential sampling is preferable to quadratic width and linear depth. This trade-off is real and acknowledged, and a deeper analysis belongs to future work when hardware capabilities evolve.
>
>  _...assumption that a unique Condorcet winner will emerge reliably...:_ A Condorcet winner emerges reliably under maximum randomness; hardware noise and class difficulty are separate concerns.
>
> _...discuss scenarios where majority voting may fail...:_ We address this explicitly—inputs without a unique winner are classified as “unresolvable.” Our approach discards these cases rather than forcing an arbitrary decision.
>
>
> **Impact on Practicality:** Please see our general comment on Scope Clarification.
>
> **Trade-off Concerns:** Please see our general comment on Quadratic vs Exponential Scaling. Note that our method does not introduce any exponentially increasing qubit requirement.
>
> **Incomplete Experiments:* This is a fair point. We recently optimized our post-processing pipeline but lacked sufficient HPC resources to rerun over 1{,}000 training and inference runs before the deadline. These results will be included in the camera-ready version as our resource grants reset.
>
> **Baseline Comparisons:**
>     _...more recent and sophisticated quantum multiclass classifier encoding schemes...:_ Our contribution focuses on output encoding strategies for VQCs under single-shot inference constraints. Methods such as amplitude encoding or mixed-state discriminators involve fundamentally different design choices—data embedding, circuit depth, and hybridization—that are orthogonal to our scope. Comparing these would require a separate study on ansatz-level architectures, which we will note as future work.
>
>    _Classical Comparisons:_ While classical one-vs-one frameworks share the conceptual idea of pairwise decomposition, our work does not claim quantum advantage over classical models. Our goal is to address the sampling bottleneck specific to quantum classifiers, not to outperform classical baselines. We will clarify this distinction and explicitly state that our contribution is complementary, not competitive.
>
> **Theoretical Justification:**
>     _Shot Resolvability:_ This metric is not intended as a fundamental theorem but as a practical measure of inference reliability under limited sampling. Its behavior follows from the combinatorial structure of encodings: one-hot validity decays exponentially with $K$, binary encodings suffer from Hamming errors, and tournament aggregation converges to a unique winner with high probability (Malinovsky \& Moon, 2024).
>     _Entanglement and Error Propagation:_ Our method preserves global entanglement, but analyzing error propagation under hardware noise is beyond the scope of this work. We will state this limitation explicitly and note it as future work.
>
> **Application and Realism:** Please see our general comment on Realism, Imbalance, and Overlap.

---

### Author Response · Authors · 2025-11-21
**General Statement**

**Scope Clarification**
Our work does not claim a breakthrough for NISQ hardware. All results assume best-case conditions under noiseless simulation, applied equally to all methods. This isolates algorithmic behavior from hardware-specific noise and focuses on the theoretical scaling properties of output encodings.
We explicitly assume a future outlook where quantum computers continue to grow in qubit count—similar to historical trends—and where noise either remains constant or decreases through error correction or hardware improvements. These assumptions are standard in theoretical QML research and are clearly stated in the paper. We have discovered and corrected several statements in the manuscript that may have given the wrong impression in this regard. We offer to provide our notes about these changes upon request to keep this response concise, noting that OpenReview provides an even more thorough pdf-diff.

**Noise and Hardware Robustness**
We distinguish between two notions of noise in quantum computing: (i) intrinsic quantum randomness, which arises from the probabilistic nature of measurement and motivates our focus on single-shot inference and resolvability, and (ii) hardware-level noise, such as decoherence, thermal fluctuations, and cross-talk, which stem from the physical limitations of current devices. Our contributions address (i) under idealized conditions while (ii) remains an open challenge and is outside the scope of this work. We clarify this as best as possible in the revised manuscript.

We acknowledge that robustness to hardware noise is an open challenge. While Appendix A.1.2 includes inference under IBM Qiskit noise models, a full systematic study of noise resilience is beyond the scope of this work. Importantly, our claims about tournament encoding hold under worst-case tournament conditions and do not rely on noise assumptions.
Regarding Condorcet cycles: theory guarantees that the probability of a unique winner converges to unity as $K \to \infty$ (Malinovsky \& Moon, 2024). We view deeper exploration of tournament theory—such as tie-breaking strategies and cycle analysis—as an exciting extension for future work. Given the breadth of this field, a comprehensive treatment is beyond the scope of a single paper, but we will highlight these directions explicitly in the revision.

**Quadratic vs Exponential Scaling**
We freely acknowledge that tournament encoding requires $O(K^2)$ qubits, which is a limitation for large $K$. However, this polynomial growth is fundamentally different from the exponential shot requirements inherent to one-hot encoding and the irreducible sampling overhead of expectation-based methods.
Our contribution introduces a complexity trade-off: quadratic resource scaling (potentially reducible via sparse tournaments or hierarchical schemes, etc.) for tournament inference versus exponential sampling cost growth for the alternatives (irreducible). No one in the field can make definitive claims about whether qubit count or shot requirements will dominate in the future, but we argue that reducible > irreducible and quadratic < exponential, making this a worthwhile research direction.
We will add a resource scaling analysis and explicitly frame this trade-off in the revision.

**Hardware Experiments**
We respect the request for real-device experiments but note that training thousands of PQCs on NISQ hardware is computationally expensive and ethically questionable given the resource cost (e.g., helium lossage) for toy datasets. Our primary contribution—the theoretical link between tournament winners and shot resolvability—would be obscured by hardware noise, which affects all methods equally.
We will clarify this rationale and emphasize that hardware validation is future work.

**Realism, Imbalance, and Overlap**

_Class Imbalance \& Semantic Overlap:_ We agree that imbalance and overlapping class boundaries can influence tournament aggregation. These effects are not unique to quantum classifiers—they also affect classical one-vs-one frameworks. Our current study assumes balanced datasets to evaluate encoding strategies under controlled conditions. We will explicitly state this assumption and note that extending the framework to handle imbalance (e.g., weighted voting, tie-breaking rules) and ambiguous boundaries is an important direction for future work.

_Real-World Scaling:_ We do not claim applicability to large-scale real-world datasets or NISQ hardware at this stage. All experiments use standard benchmark subsets (MNIST Digits and FashionMNIST) to isolate algorithmic behavior without conflating results with hardware limitations or dataset-specific complexities. Applying this method to high-dimensional, large-class datasets or real hardware requires addressing quadratic resource scaling and noise resilience—both acknowledged limitations in Section 5. We will highlight these constraints more prominently and frame them as future research directions.

---

### Author Response · Authors · 2025-12-03

**Final Comments**

We thank the AC for managing the challenges caused by the recent leak and for ensuring a fair review process. Below we summarize the major revisions incorporated into the Rebuttal Revisions in response to earlier feedback:

*Introduction* - The opening paragraph was revised for accuracy and clarity. We added explicit statements on scope and limitations throughout the paper and expanded the reference list to strengthen context.

*Methods* - Section 3.2 now includes an illustrative tournament aggregation example (Figure~2) and clearer explanations of majority-based Condorcet aggregation. Additional clarifications on scope and limitations were integrated where relevant.

*Results* - The updated version includes complete data for $K=6$ across all experiments. Tables and Figure~3 have been revised accordingly, and prior notes about missing data have been removed.

*Limitations* - This section was substantially expanded to clarify the quadratic scaling trade-off and its relationship to exponential sampling costs. We outline future directions for mitigating scaling issues, including hierarchical and sparse tournaments, hybrid aggregation schemes, and extensions from tournament theory. We also explicitly state that all experiments assume noiseless simulation and balanced datasets, and discuss the implications of class imbalance and semantic overlap as open challenges.

*General* -

 - Careful review and correction of misleading terminology, particularly the use of “noise,” which previously risked conflating intrinsic quantum randomness with hardware-level noise.

 - Multiple minor clarifications to prevent misinterpretation of scope and intent.

---

### Meta-Review · Area_Chair_8Rty · 2025-12-27

**Summary:**

In this paper, the authors proposes an encoding framework, named Quan-Dorcet, for quantum multiclass classification. The method is based on pairwise round-robin tournament comparisons, aiming to improve the accuracy and resolvability of single-shot and few-shot inference.

In the reviews, there are general concerns about:
- Scalability: The required number of qubits scales quadratically with the number of classes.
- Lacking QPU execution: No experiments on real quantum devices.
- Theoretical justifications.

In the authors' reply, points are get clarified but these issues still exist: on the theory side there's no rigorous justification about the advantage of the proposed method, and on the experimental side it has stability issue and there's no experiment on real quantum devices. These make the paper not reaching the high standard of ICLR. Considering the overall concerns raised from the review, the decision by this meta-review is rejection.

**Reviewer Concerns:**

Please see the message above - there are still outstanding issues about scalability, real-world experiments, and theoretical justificationss.

**Reviewer Scores:**

I don't think the scores will notably change if reviewers were able to participate fully in the discussion. There were discussions with Reviewers JMzA and BZ68, who both kept the score of 4. The other score 4 review by Reviewer XFm6 is a very detailed one and the raised issues were justified. The only positive review of score 6 is relatively concise.

---

### Decision · Program_Chairs · 2026-01-26

Reject